# Time elapsed between Zika and dengue virus infections affects antibody and T cell responses

Erick X. Pérez-Guzmán[1,9], Petraleigh Pantoja[1,2], Crisanta Serrano-Collazo[1], Mariah A. Hassert[3], Alexandra Ortiz-Rosa [4], Idia V. Rodríguez[2], Luis Giavedoni[5], Vida Hodara[5], Laura Parodi[5], Lorna Cruz[1,2], Teresa Arana[1,2], Laura J. White[6], Melween I. Martínez[1,2], Daniela Weiskopf[7], James D. Brien[3], Aravinda de Silva[6], Amelia K. Pinto [3] & Carlos A. Sariol [1,2,8]

Zika virus (ZIKV) and dengue virus (DENV) are co-endemic in many parts of the world, but the impact of ZIKV infection on subsequent DENV infection is not well understood. Here we show in rhesus macaques that the time elapsed after ZIKV infection affects the immune response to DENV infection. We show that previous ZIKV exposure increases the magnitude of the antibody and T cell responses against DENV. The time interval between ZIKV and subsequent DENV infection further affects the immune response. A mid-convalescent period of 10 months after ZIKV infection results in higher and more durable antibody and T cell responses to DENV infection than a short period of 2 months. In contrast, previous ZIKV infection does not affect DENV viremia or pro-inflammatory status. Collectively, we find no evidence of a detrimental effect of ZIKV immunity in a subsequent DENV infection. This supports the implementation of ZIKV vaccines that could also boost immunity against future DENV epidemics.

[1] Department of Microbiology and Medical Zoology, University of Puerto Rico-Medical Sciences Campus, San Juan, PR, USA. [2] Unit of Comparative Medicine, Caribbean Primate Research Center and Animal Resources Center, University of Puerto Rico-Medical Sciences Campus, San Juan, PR, USA. [3] Department of Molecular Microbiology & Immunology, Saint Louis University School of Medicine, Saint Louis, MO, USA. [4] Department of Biology, University of Puerto Rico-Río Piedras Campus, San Juan, PR, USA. [5] Texas Biomedical Research Institute, San Antonio, TX, USA. [6] Departments of Microbiology & Immunology, University of North Carolina-Chapel Hill, Chapel Hill, NC, USA. [7] Division of Vaccine Discovery, La Jolla Institute for Immunology, La Jolla, CA, USA. [8] Department of Internal Medicine, University of Puerto Rico-Medical Sciences Campus, San Juan 00936 PR, USA. [9]Present address: Takeda Vaccines Inc, Cambridge, MA, USA. Correspondence and requests for materials should be addressed to C.A.S. (email: carlos.sariol@upr.edu)

Zika virus (ZIKV) is a re-emerging mosquito-borne *Flavivirus* that has recently spread in the Americas[1], and is associated with severe neurological sequelae[2–4]. ZIKV established itself in tropical and sub-tropical regions that are endemic to other closely related flaviviruses such as dengue virus (DENV). Both viruses belong to the Flaviviridae family and are transmitted by *Aedes spp.* mosquitoes. DENV is a global public health threat, having two-thirds of world's population at risk of infection, causing ~390 million infections annually[5,6]. DENV exists as four genetically similar but antigenically different serotypes (DENV1-4)[7]. Exposure to one DENV serotype confers long-lived immunity against a homotypic secondary infection. However, secondary infection with a heterologous serotype of DENV is the major risk factor to induce severe DENV disease[8–10].

Due to the antigenic similarities between DENV and ZIKV, concerns have been raised regarding the impact of DENV–ZIKV cross-reactive immunity on the development of severe clinical manifestations[11,12]. It has been demonstrated that DENV-immune sera from humans can enhance ZIKV infection in vitro[13,14], and in vivo in immune-deficient mouse models[15]. However, recent results from our group and others have shown that previous flavivirus exposure—including DENV—may have no detrimental impact on ZIKV infection in vivo in non-human primates (NHP)[16,17] and humans[18]. Moreover, these studies and others suggest that previous DENV immunity may play a protective role during ZIKV infection involving humoral and cellular responses[19–23]. On the other hand, little is known about the opposite scenario, the role of a previous ZIKV exposure on subsequent DENV infection, which is relevant to anticipate the dynamics of forthcoming DENV epidemics.

The recent ZIKV epidemic in the Americas resulted in the development of a herd immunity that may have an impact in subsequent infections with other actively circulating flaviviruses such as DENV. Thus, human subpopulations such as newborns, international travelers from non-flavivirus endemic areas, or DENV-naive subjects could be exposed to a ZIKV infection prior to DENV—since DENV declined in the Americas during ZIKV epidemic[24]. After the epidemic, herd immunity reduced ZIKV transmission, and DENV will re-emerge and potentially infect these DENV-naive ZIKV-immune subpopulations in the Americas or potentially in other geographic areas newly at risk[25,26]. An epidemiological study based on active DENV surveillance in Salvador, Brazil, suggests that the reduction of DENV cases after the ZIKV epidemic is due to protection from cross-reactive immune responses between these viruses[27]. Prospective experimental studies are needed to confirm this hypothesis. NHPs provide advantages such as an immune response comparable with humans, and the normalization of age, sex, injection route, viral inoculum, and timing of infection[28]. Although clinical manifestations by flaviviral infections are limited in NHPs[29], they have been widely used as an advanced animal model for the study of DENV and ZIKV immune response, pathogenesis, and vaccine development[16,17,28,30–33].

ZIKV antibodies (Abs) are capable of enhancing DENV infection in vitro[34]. Characterization of the specificity of DENV and ZIKV cross-reactive response revealed that ZIKV monoclonal Abs and maternally acquired ZIKV Abs can increase DENV severity and viral burden in immune-deficient mouse models[35,36]. *George et al.* showed that an early convalescence to ZIKV induced a significant higher peak of DENV viremia and a pro-inflammatory profile compared with ZIKV-naive status in rhesus macaques[37]. A recent NHPs study showed that clinical and laboratory parameters of ZIKV-immune animals were not associated with an enhancement of DENV-2 infection. However, a higher peak of DENV-2 plasma RNAemia in ZIKV-immune animals was observed compared with DENV-2 serum RNAemia

loads in control animals, but the use of different sample types may account for these differences[38]. Despite these findings, further studies are needed to dissect the complementary role of the innate, humoral, and cellular immune response to mechanistically explain these findings. Particularly, there is no evidence of the modulation and functionality of the T-cell response in the ZIKV–DENV scenario. Available studies rely upon pathogenesis and antibody studies, but there is no documented evidence as to whether cell-mediated immunity (CMI)—specifically the functional response of T cells—is modulated in a subsequent DENV infection by the presence of ZIKV immune memory.

Shorter time interval between DENV infections result in a sub-clinical secondary infection, while symptomatic secondary infections and severe DENV cases have been related with longer periods between infections[39–42]. These findings suggest that high titers of cross-reactive Abs play a time-dependent protective role between heterotypic DENV infections. Despite this evidence from DENV sequential infections, it remains poorly understood if the same applies to the time interval between ZIKV–DENV sequential infections. So currently, the role of multiple convalescent periods to ZIKV in the outcome of DENV and other flavivirus infections is in the forefront of discussions based on the limited studies available in experimental models and a lack of characterized human prospective cohorts of this scenario yet[27,43–45].

The objective of our study is to investigate the immune modulatory role of an early convalescence and middle convalescence after ZIKV infection on the outcome of a subsequent DENV infection in rhesus macaques. We infect NHP cohorts who were ZIKV immune for 10 months (middle convalescence), 2 months (early convalescence) or naive for ZIKV with DENV. The 2 months cohort is selected for direct comparison with previous work in NHP[37], while the 10 months cohort is selected to test a longer period of convalescence to ZIKV, where the cross-reactive Abs are known to wane[46]. In each of these groups, we assess DENV pathogenesis, the elicited Ab response, and characterize the CMI. This study provides evidence that the presence of ZIKV immune memory contributes to increase the magnitude of the immune response—more efficient after longer ZIKV pre-exposure—against a DENV infection, without promoting enhancement of DENV viremia nor inducing higher levels of pro-inflammatory cytokines.

## Results

**DENV challenge and clinical status of rhesus macaques.** The experimental design includes three cohorts of rhesus macaques (*Macaca mulatta*), within the age range considered as young adults (Supplementary Fig. 1k), that were challenged with DENV-2 (NGC-44 strain), monitored, and bled for 3 months (Fig. 1). Two cohorts were previously exposed to ZIKV: cohort 1 (ZIKVPF-10mo) was comprised of four animals that had been exposed to ZIKV H/PF/2013 strain 10 months before DENV-2 challenge (mid convalescence), and cohort 2 (ZIKVPR-2mo) comprised of six animals that had been exposed to ZIKV PRVABC59 strain 2 months before DENV-2 challenge (early convalescence). Both ZIKV strains used for previous exposure of these groups are > 99.99% comparable in amino acid identity (Supplementary Table 1). An additional cohort 3 (naive) included four animals naive to ZIKV/DENV as a control group. After DENV challenge, all macaques were extensively monitored, and sample collection was performed at various timepoints up to 90 days post infection (dpi) for serum and PBMCs isolation.

Vital signs such as weight (kg) and temperature (°C) were monitored. Also, complete blood cell counts (CBC) and comprehensive metabolic panel (CMP) were performed before (baseline: day 0) and after DENV infection at multiple timepoints

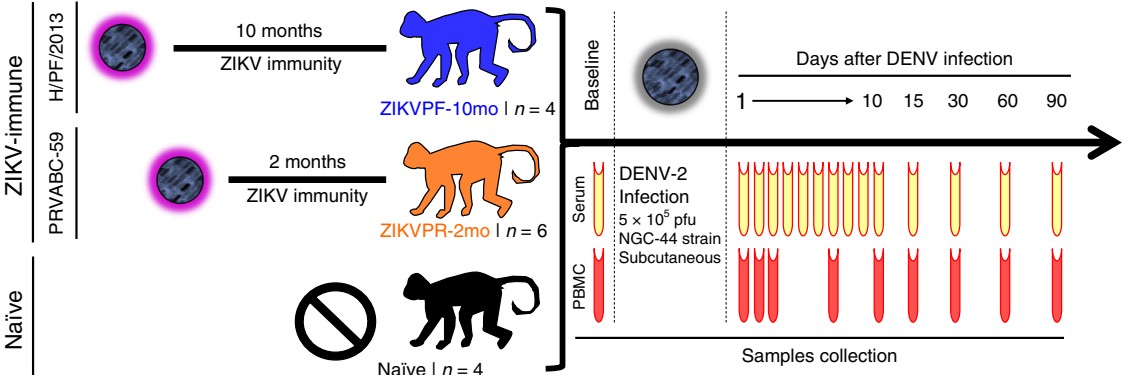

**Fig. 1** Experimental design for DENV-2 challenge of ZIKV-immune and naive macaques. Fourteen young adult male rhesus macaques (*Macaca mulatta*), matched in age and weight, were divided into three cohorts. ZIKVPF-10mo (*n* = 4): composed of four animals (5K6, CB52, 2K2, and 6N1) that were inoculated with 1×10⁶ pfu/500 µl of the ZIKV H/PF/2013 strain subcutaneously 10 months before (middle convalescence) DENV-2 challenge. ZIKVPR-2mo (*n* = 6): composed of six animals (MA067, MA068, BZ34, MA141, MA143, and MA085) that were inoculated with 1×10⁶ pfu/500 µl of the contemporary ZIKV PRVABC59 strain 2 months before (early convalescence) DENV-2 challenge. Both ZIKV strains used for previous exposure of these groups are > 99.99% comparable in amino acid identity (Supplementary Table 1). Naive (*n* = 4): composed of four ZIKV/DENV naive animals (MA123, MA023, MA029, and MA062) as a control group. Prior to DENV-2 challenge, all animals were subjected to quarantine period. All cohorts challenged subcutaneously (deltoid area) with 5 × 10⁵ pfu/500 µl of DENV-2 New Guinea 44 strain (NGC44). After DENV-2 challenge, all animals were extensively monitored for evidence of disease and clinical status by vital signs, such as external temperature (°C), weight (Kg), CBC, and CMP panels at the Caribbean Primate Research Center (CPRC). Blood samples were collected at baseline, 1–10, 15, 30, 60, and 90 days after DENV infection. In all timepoints, the blood samples were used for serum separation (yellow). PBMCs isolation (red) was performed in different tubes with citrate as anticoagulant at baseline, 1, 2, 3, 7, 10, 15, 30, 60, and 90 days after DENV infection

(CBC: 0, 7, 15 dpi; CMP: 0, 7, 15, 30 dpi). Neither symptomatic manifestations nor significant differences in weight or temperature were observed in any of the animals after DENV infection up to 90 dpi (Supplementary Fig. 1a, b). Likewise, no significant differences between groups were detected in CBC parameters: white blood cells (WBC), lymphocytes (LYM), neutrophils (NEU), monocytes (MON), and platelets (PLT) after DENV infection compared with basal levels of each group (Supplementary Fig. 1c–g). CMP levels showed no differences in alkaline phosphatase and aspartate transaminase (AST) (Supplementary Fig. 1h–i). Although within the normal range, levels of alanine transaminase (ALT) were significantly higher in the ZIKVPR-2mo group compared with its baseline at 7 dpi ($p = 0.0379$, two-way ANOVA Dunnett test), but at 15 and 30 dpi values returned to baseline levels (Supplementary Fig. 1j). Overall, except for the isolated increase of ALT at 7 dpi in ZIKVPR-2mo, the clinical profile suggests that the presence of ZIKV-immunity did not significantly influence the clinical outcome of DENV infection.

**DENV RNAemia not enhanced by previous ZIKV immunity**. RNAemia levels in NHPs serum were quantified by qRT-PCR at baseline, 1–10, and 15 dpi to determine if the presence of early convalescence (ZIKVPR-2mo) or mid convalescence (ZIKVPF-10mo) to ZIKV alters DENV RNAemia kinetics. No significant differences between groups were observed in detected levels of DENV genome copies per milliliter of serum overtime (Fig. 2a). We noted that in the ZIKVPF-10mo group, three out of four animals were able to keep the RNAemia level below $10^3$ genome copies the next day after DENV infection. This group started an early clearance of the RNAemia at 7 dpi, with only one out of four animals having detectable levels by days 8 and 9 pi. For ZIKVPR-2mo and naive animals, the clearance of detectable RNAemia started at 8 dpi, in four out of six and one out of four of the animals, respectively. Naive animals had the most delayed clearance of RNAemia with at least half of the animals with detectable levels of viral RNA until day 9 pi. RNAemia was completely resolved in all animals by 10 dpi. In summary, ZIKVPF-10mo had 7.25, ZIKVPR-2mo 7.5, and naive animals 8

mean days of detectable RNAemia after DENV infection (Fig. 2b). In addition, the area under the curve (AUC) was calculated, but no statistically significance differences were observed in the RNAemia peak among groups (Supplementary Fig. 2). However, the AUC trend to be lower in both ZIKV-immune groups. In terms of the kinetics, a delay in the peak RNAemia set point was observed in both ZIKV-immune groups (switch from day 2 to days 5 and 6) followed by higher, but non-significant, levels compared with the naive group, and a subsequent early RNAemia clearance in both ZIKV-immune groups. Together these results show that, although no statistically significant differences among groups were observed, previous immunity to ZIKV is not associated with an increase in DENV RNAemia; even more, a mid-convalescence to ZIKV tended to develop a shorter viremic period.

**Pro-inflammatory cytokines not exacerbated by ZIKV immunity**. To determine if the characterized cytokine profile of an acute DENV infection was modulated by ZIKV immunity, we assessed the serum concentration (pg/ml) of eight cytokines/chemokines by Luminex multiplex at baseline, 1, 2, 3, 5, 10, 15, and 30 dpi. The naive group showed significant higher levels of Type I interferon alpha (IFN-α) and pro-inflammatory cytokines such as Interleukin-6 (IL-6), and monokine induced by IFN-gamma (MIG/CXCL9) (Fig. 3a–c). IFN-α was highest at 5 dpi (Fig. 3a: $p < 0.0001$ vs ZIKVPF-10mo and $p = 0.0003$ vs ZIKVPR-2mo, two-way ANOVA Tukey test). IFN-α has been demonstrated to be involved in the innate antiviral immunity, and elevated levels are associated with higher viral load and antigen availability. IL-6, a multifunctional cytokine involved in immune response regulation and many inflammatory reactions showed the highest levels at 1 dpi in naive animals (Fig. 3b: $p = 0.0115$ vs ZIKVPF-10mo and $p = 0.0185$ vs ZIKVPR-2mo, two-way ANOVA Tukey test). Finally, MIG/CXCL9, which is a potent chemoattractant involved in leukocyte trafficking demonstrated the highest levels at 10 dpi in naive animals (Fig. 3c: $p = 0.0004$ vs ZIKVPR-2mo, two-way ANOVA Tukey test). On the other hand, the mid-convalescent ZIKVPF-10mo group showed higher levels

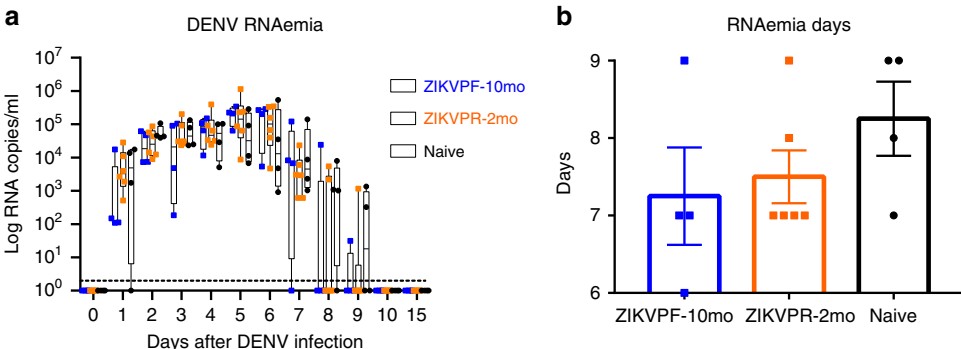

**Fig. 2** Previous ZIKV immunity does not contribute to an increase of DENV RNAemia. **a** DENV-2 RNA kinetics in ZIKV-immune and naive animals at baseline, day 1–10, and day 15 after DENV infection. RNA genome copies (Log10) per milliliter of serum were measured by qRT-PCR. Symbols represent individual animals per cohort: blue squares (ZIKVPF-10mo), orange squares (ZIKVPR-2mo), and black circles (naive). Box and whiskers show the distribution of log-transformed values per group per timepoint. Boxes include the mean value per group, while whiskers depict the minimum and maximum values for each group. Cutted line mark the limit of detection (20 genomes copies). Statistically significant differences between groups were determined using two-way ANOVA adjusted for Tukey's multiple comparisons test including 12 families, and 3 comparisons per family. **b** The total days that DENV-2 RNAemia was detected for each animal within cohorts. Bars represent mean days per cohort. Source data are provided as a Source Data file

of CXCL10 (IP-10) (Fig. 3g) at day 1 ($p = 0.0198$ vs ZIKVPR-2mo, two-way ANOVA Tukey test), 5 ($p = 0.0487$ vs naive, two-way ANOVA Tukey test), and 10 pi ($p = 0.0009$ vs ZIKVPR-2mo, two-way ANOVA Tukey test). CXCL10 is a T-cell activating chemokine and chemoattractant for many other immune cells. Also, this group showed higher levels of perforin (Fig. 3h) at day 10 ($p = 0.0024$ vs naive and $p = 0.0190$ vs ZIKVPR-2mo, two-way ANOVA Tukey test) and 15 pi ($p = 0.0178$ vs naive, two-way ANOVA Tukey test). Perforin is an effector cytolytic protein released by activated cytotoxic CD8 + T cells and natural killer (NK) cells. No significant differences between groups were observed for other pro-inflammatory citokines, such as monocyte chemoattractant protein 1 (MCP-1), macrophage inflammatory protein 1-beta (MIP-1β), and IL-1 receptor antagonist (IL-1RA) (Fig. 3d–f). Collectively, these results demonstrate that the presence of ZIKV immunity does not exacerbate pro-inflammatory status after DENV infection while mid-convalescence immunity to ZIKV stimulated levels of mediators mainly involved in the activation of cell-mediated immune response.

**DENV and ZIKV cross-reactive antibody response**. An ELISA-based serological profile was performed to determine the contribution of ZIKV immunity in the cross-reactive Ab response before and after DENV infection. We assessed the levels of DENV IgM and IgG, and cross-reactivity with ZIKV (IgM, IgG, NS1-IgG, and EDIII–IgG) at multiple timepoints (Supplementary Fig. 3). Naive cohort showed a significant higher peak of IgM (Supplementary Fig. 3a) characteristic of a primary DENV infection at 15 and 30 dpi ($p < 0.0001$ vs ZIKVPF-10mo and $p = 0.0004$ vs ZIKVPR-2mo, $p = 0.0044$ vs ZIKVPF-10mo and $p = 0.0179$ vs ZIKVPR-2mo, respectively, two-way ANOVA Tukey test). This indicates the productive and acute DENV infection, while ZIKV immune groups showed lower levels of IgM resembling a heterotypic secondary infection. The total DENV IgG levels (Supplementary Fig. 3b) of both ZIKV-immune groups were significantly higher compared with naive since baseline (cross-reactive ZIKV–IgG Abs) and 7, 15, 30, 60, and 90 (the latter for ZIKVPF-10mo only) (ZIKVPF-10mo vs naive: $p = 0.0010$, $p < 0.0001$, $p < 0.0001$, $p < 0.0001$, $p < 0.0001$, $p = 0.0016$; ZIKVPF-2mo vs naive: $p = 0.0029$, $p = 0.0002$, $p < 0.0001$, $p < 0.0001$, $p = 0.0006$; two-way ANOVA Tukey test). The ZIKVPF-10mo group showed significant higher levels than ZIKVPR-2mo group at 30 and 90 dpi ($p = 0.0242$ and $p = 0.0348$, two-way

ANOVA Tukey test). Overall, ZIKVPF-10mo developed higher and long-lasting levels of DENV IgG.

In contrast, ZIKV IgM levels were under or near the limit of detection in all groups over time after DENV infection, despite several significant differences between groups (Supplementary Fig. 3c). ZIKV IgG levels (Supplementary Fig. 3d) were high in both ZIKV-immune groups at baseline and 7 dpi compared with naive ($p < 0.0001$ vs naive, two-way ANOVA Tukey test), suggesting that although different pre-infecting ZIKV strains, the previous elicited IgG response against both ZIKV strains is comparable. After DENV infection, an increase of ZIKV IgG was shown and remain constantly high at 15, 30, 60, and 90 dpi in both ZIKV-immune groups ($p < 0.0001$ vs naive for all timepoints, two-way ANOVA Tukey test), suggesting that DENV has the potential to stimulate ZIKV-binding Ab-producing plasmablasts. In addition, to elucidate the composition of similar ZIKV IgG levels in ZIKV-immune groups, we measured ZIKV-specific NS1 IgG (Supplementary Fig. 3e) and ZIKV-specific EDIII IgG (Supplementary Fig. 3f) levels. Although ZIKVPR-2mo showed significant differences compared with naive at 30, 60, and 90 dpi ($p < 0.0001$, $p = 0.0001$, $p = 0.0159$; two-way ANOVA Tukey test), we observed a significantly higher expansion and long-lasting response of ZIKV NS1-specific Abs in the ZIKVPF-10mo group compared with the ZIKVPR-2mo group at baseline, 60 and 90 dpi ($p = 0.0036$, $p = 0.0071$, $p = 0.0294$; two-way ANOVA Tukey test) and also compared with naive animals at all timepoints ($p < 0.0001$, two-way ANOVA Tukey test). Moreover, higher magnitude of ZIKV-specific EDIII–IgG levels in the ZIKVPF-10mo group than in the ZIKVPR-2mo group was observed compared to naïve at baseline (ZIKVPF-10mo only), 15, 30, and 60 (ZIKVPF-10mo vs naive: $p = 0.0092$, $p < 0.0001$, $p < 0.0001$, $p = 0.0034$; ZIKVPR-2mo vs naive: $p = 0.0003$, $p = 0.0014$, $p = 0.0055$; two-way ANOVA Tukey test), suggesting that a ZIKV mid-convalescence promotes an expansion of higher magnitude of ZIKV EDIII–IgG Abs from ZIKV memory B cells (MBC). However, those higher cross-reacting levels decrease overtime as expected. In summary, a boost of DENV and ZIKV Abs is triggered by the presence of ZIKV immunity, and the expansion of specific- and cross-reactive Abs is higher on magnitude and durability when a mid-convalescence immunity to ZIKV is present.

**DENV neutralization is boosted by ZIKV immunity**. Neutralizing antibodies (NAbs) are essential to combat DENV and ZIKV infection. The maturation and potency of this response is known to define to a great extent the infection outcome[11,47].

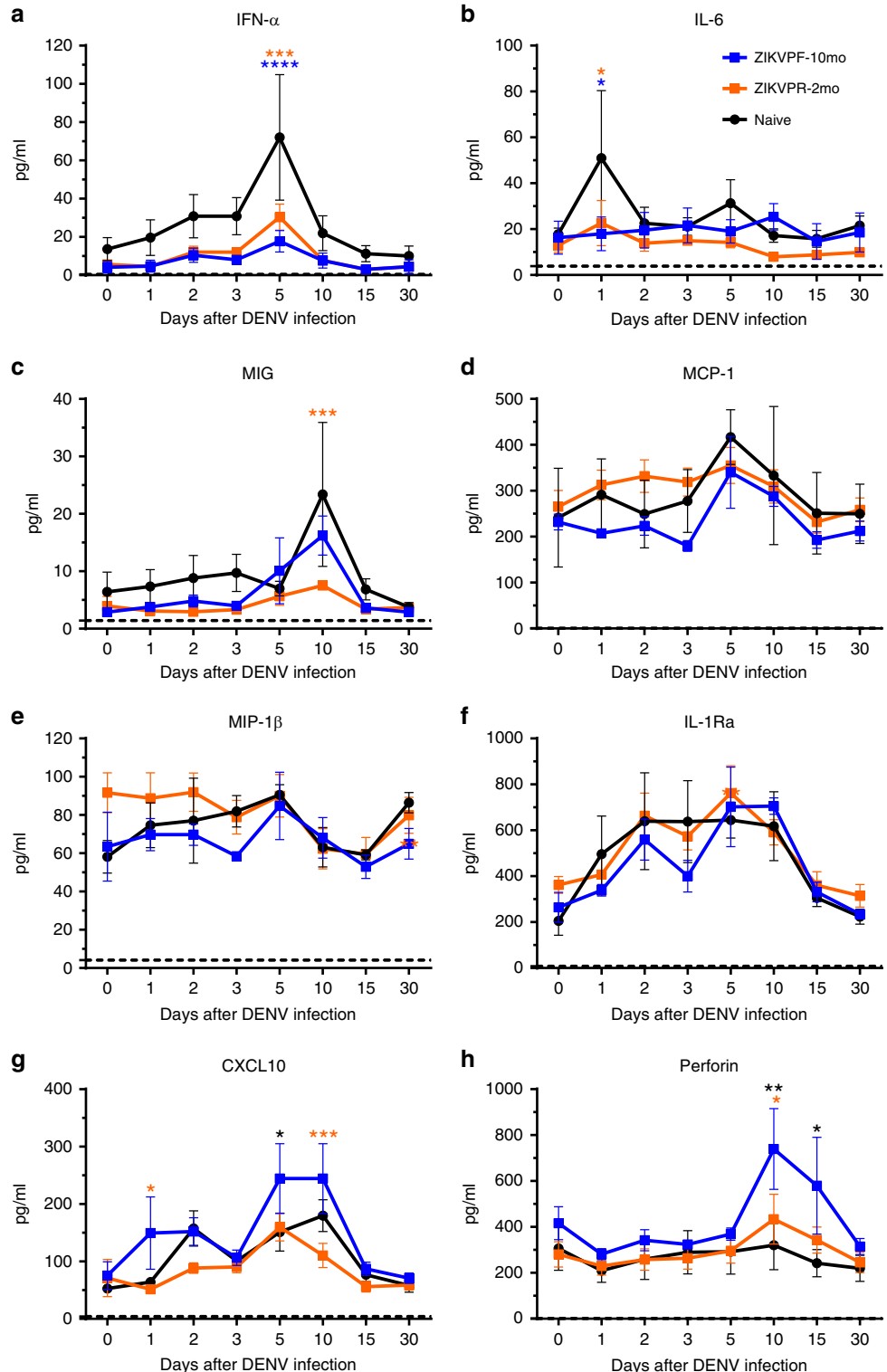

**Fig. 3** ZIKV immunity does not exacerbate levels of pro-inflammatory cytokines. Cytokines and chemokines expression levels were determined in serum (pg/ml) by multiplex bead assay (Luminex) at baseline, 1, 2, 3, 5, 10, 15, and 30 days after DENV infection. The panel includes: **a** interferon alpha (IFN-α), **b** interleukin-6 (IL-6), **c** monokine induced by IFN-gamma (MIG/CXCL9), **d** monocyte chemoattractant protein 1 (MCP-1/CCL2), **e** macrophage inflammatory protein 1-beta (MIP-1β/CCL4), **f** IL-1 receptor antagonist (IL-1RA), **g** C–X–C motif chemokine 10 (CXCL10/IP-10), and **h** perforin. Symbols connected with lines represent mean expression levels detected of each cytokine/chemokine per cohort over time: blue squares (ZIKVPF-10mo), orange squares (ZIKVPR-2mo), and black circles (naive). Error bars indicate the standard error of the mean (SEM) for each cohort per timepoint. Cutted line marks the limit of detection for each individual cytokine/chemokine. Statistically significant differences between groups were calculated using two-way ANOVA adjusted for Tukey's multiple comparisons test including eight families, and three comparisons per family. Significant multiplicity adjusted p-values (*< 0.05, **< 0.01, ***<0.001, ****< 0.0001) are shown colored representing the cohort against that particular point where is a statistically significant difference between groups. Source data are provided as a Source Data file

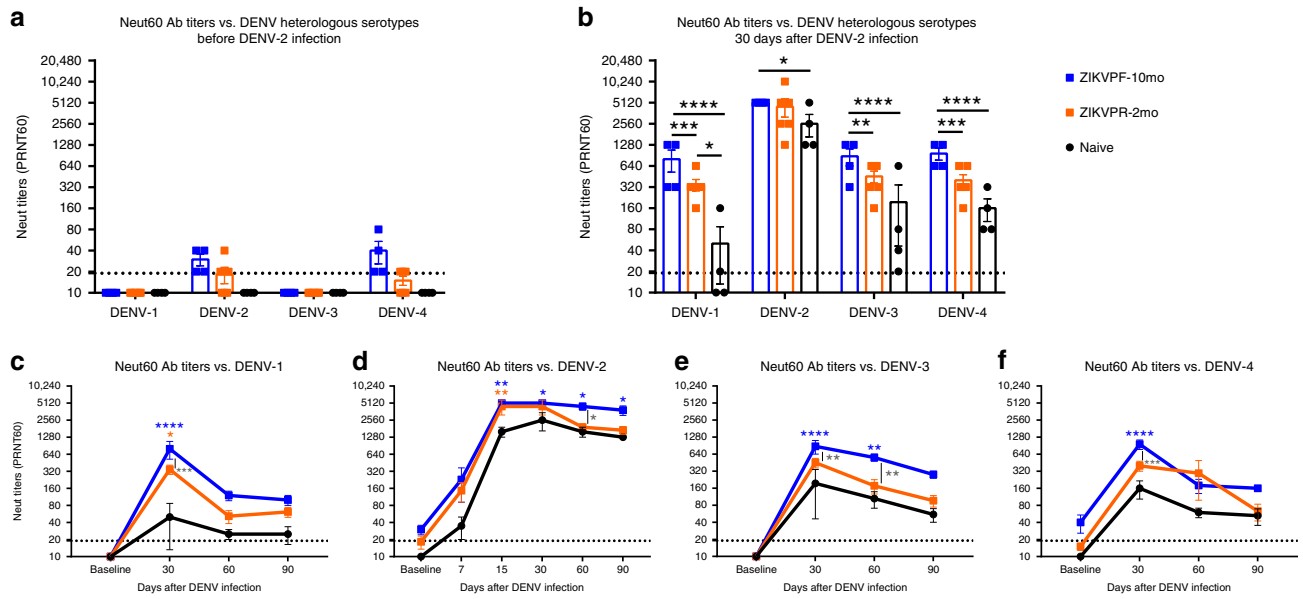

**Fig. 4** Neutralization of DENV serotypes by ZIKV-immune animals is higher in magnitude. The magnitude of the neutralizing antibody (NAb) response was determined (**a**) before and (**b**) 30 days after DENV infection by plaque reduction neutralization test (PRNT) against all DENV serotypes. **c–f** The durability of the neutralizing response was assessed measuring NAb titers up to 90 dpi against all DENV serotypes. Symbols connected with full lines indicate mean levels of NAb titers detected per cohort over time: blue squares (ZIKVPF-10mo), orange squares (ZIKVPR-2mo), and black circles (naive). Error bars represent the standard error of the mean (SEM). PRNT60: NAb titer capable of reduce 60% or more of DENV serotypes plaque-forming units (pfu) compared with the mock (control of virus without serum). A PRNT60 1:20 titer was considered positive, and < 1:20 as a negative Neut titer. Dotted line mark < 1:20 for the negative results. Non-neutralizing titers (< 1:20) were assigned with one-half of the limit of detection for graphs visualization (1:10). Statistically significant differences between groups were calculated using two-way ANOVA adjusted for Tukey's multiple comparisons test including four and six families for heterologous serotypes and DENV-2, respectively, and three comparisons per family. Significant multiplicity adjusted $p$-values (*< 0.05, **< 0.01, ***< 0.001, ****< 0.0001) are shown. Blue and orange asterisks represent significant difference between the corresponded ZIKV immune groups and naive group, and gray asterisks indicate a significant difference between ZIKV immune groups. Source data are provided as a Source Data file

Accordingly, we tested the neutralization capacity of NAbs in serum from ZIKV-immune and naive animals before and after DENV infection, to determine whether an early convalescence or mid convalescence to ZIKV affected the NAb response. Plaque reduction neutralization test (PRNT) was performed to elucidate the NAb titers of all groups against all DENV serotypes and both ZIKV pre-infecting strains. Before infection with DENV, the naive groups had no detectable NAb levels (< 1:20 PRNT60 titers) against all DENV serotypes, while ZIKV-immune groups showed low cross-NAb titers against DENV-2 and DENV-4 (Fig. 4a). These cross-reactive levels were higher in the ZIKVPF-10mo group than in the ZIKVPR-2mo group for both viruses. The peak of high NAb titers occurred at 30 days after DENV infection for all groups (ZIKVPF-10mo > ZIKVPR-2mo > naive) against all DENV serotypes (DENV-2 > DENV-4 > DENV-3 > DENV-1) (Fig. 4b). The ZIKVPF-10mo group neutralized all DENV serotypes with significant higher potency than naive animals ($p$ < 0.0001, $p$ = 0.0337, $p$ < 0.0001, $p$ < 0.0001 for DENV1-4; two-way ANOVA Tukey test) and the ZIKVPR-2mo group, except for DENV-2, that both ZIKV-immune groups have comparable neutralization magnitude at 30 dpi ($p$ = 0.0002, $p$ = 0.7636, $p$ = 0.0016, $p$ = 0.0004 for DENV1-4; two-way ANOVA Tukey test). However, the neutralization kinetics by sigmoidal response curves suggest higher percent of neutralization against DENV-2 over-time in the group with mid convalescence to ZIKV (Supplementary Fig. 4). On the other hand, the ZIKVPR-2mo group showed significantly higher potency of the NAb response only against DENV-1 compared with naive animals ($p$ = 0.0146; two-way ANOVA Tukey test) (Fig. 4b).

In addition, we tested whether the NAb titers that peak at 30 dpi for all groups remain constant over time (up to 90 dpi)

against all DENV serotypes (Fig. 4c–f). In general, the neutralizing response of the ZIKVPF-10mo group maintained higher NAb titers up to 90 dpi compared with ZIKVPR-2mo and naive groups. Significant differences between ZIKVPF-10mo and ZIKVPR-2mo groups were observed against DENV-1, -3, and -4 at day 30 pi ($p$ = 0.0002, $p$ = 0.0016, $p$ = 0.0004; two-way ANOVA Tukey test) and at day 60 pi against DENV-2 and DENV-3 ($p$ = 0.0179, $p$ = 0.0047; two-way ANOVA Tukey test). The NAb response of the ZIKVPF-10mo group was even more significantly higher compared with the naive group at day 15 (only performed for the infecting serotype to monitor early neutralizing activity), day 30, 60, and 90 pi against DENV-2 ($p$ = 0.0022, $p$ = 0.0337, $p$ = 0.0146, $p$ = 0.0337; two-way ANOVA Tukey test); at day 30 pi against DENV-1 ($p$ < 0.0001, two-way ANOVA Tukey test); at day 30 and 60 pi against DENV-3 ($p$ < 0.0001, two-way ANOVA Tukey test); and at day 30 pi against DENV-4 ($p$ < 0.0001, two-way ANOVA Tukey test). In contrast, the ZIKVPR-2mo group showed a NAb response with a magnitude and long-lasting levels comparable with the naive group, except at day 15 and 30 pi against DENV-2 and DENV-1, respectively ($p$ = 0.0067, $p$ = 0.0146; two-way ANOVA Tukey test). The NAb response was long-lasting in the ZIKVPF-10mo group compared with other groups as supported by the data from days 30 and 60 p.i. At day 90 pi, although no significant differences were observed between all groups, the ZIKVPF-10mo group showed a consistent trend to maintain higher NAb titers against all DENV serotypes, indicating a higher and long-lasting breadth of cross-neutralization within DENV serocomplex.

Collectively, these results demonstrate that a mid convalescence to ZIKV provokes a boost of the magnitude and durability of the Ab response against all DENV serotypes more effectively

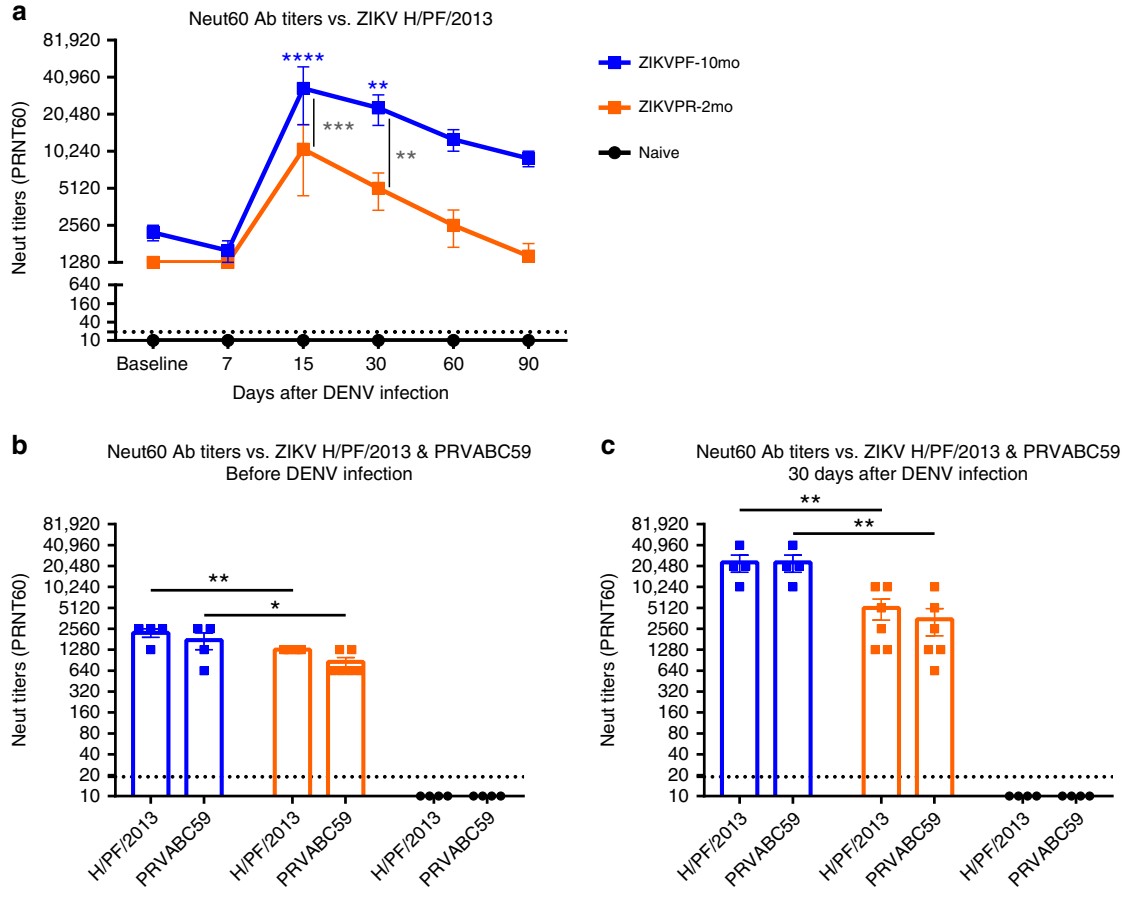

**Fig. 5** ZIKV neutralization is boosted after DENV infection, and is strain independent. **a** NAb titers against ZIKV H/PF/2013 were determined by PRNT60 at baseline, 7, 15, 30, 60, and 90 days after DENV infection. Comparison of NAb titers between pre-infecting ZIKV strains was performed (**b**) before and (**c**) after DENV infection. Symbols connected with full lines indicate mean levels of NAb titers detected per cohort over time: blue squares (ZIKVPF-10mo), orange squares (ZIKVPR-2mo), and black circles (naive). Error bars represent the standard error of the mean (SEM). PRNT60: NAb titer capable of reduce 60% or more of ZIKV strains plaque-forming units (pfu) compared with the mock (control of virus without serum). A PRNT60 1:20 titer was considered positive, and < 1:20 as a negative Neut titer. Dotted line mark < 1:20 for negative results. Non-neutralizing titers (< 1:20) were assigned with one-half of the limit of detection for graphs visualization (1:10). Statistically significant differences between groups were calculated using two-way ANOVA adjusted for Tukey's multiple comparisons test including six and two families for panel **a** and **b**, **c**, respectively, and three comparisons per family. Significant multiplicity adjusted $p$-values (*< 0.05, **< 0.01, ***< 0.001, ****< 0.0001) are shown. Blue and orange asterisks represent significant difference between the corresponded ZIKV-immune groups and naive group, and gray asterisks indicate a significant difference between ZIKV-immune groups. Source data are provided as a Source Data file

than in animals with an early convalescence to ZIKV, but also higher compared with a de novo DENV-specific NAb response of the naive animals.

**ZIKV neutralization is boosted by DENV infection.** Previous exposure to ZIKV strains in ZIKV-immune groups developed high levels of cross-reactive, non-neutralizing, and NAbs before DENV infection (baseline). To determine if this memory Ab response is strain-specific and if the difference in convalescence period to ZIKV alters the efficacy and modulation after DENV infection, we assessed the NAb levels in ZIKV-immune (ZIKVPF-10mo and ZIKVPR-2mo) and ZIKV-naive serum with both pre-infecting contemporary Asian-lineage H/PF/2013 and PRVABC59 ZIKV strains at multiple timepoints after DENV infection. At baseline, both ZIKV-immune groups showed high NAb titers against H/PF/2013 strain, which suggest that irrespective of pre-exposure to different ZIKV strains and different convalescent periods the Ab response remains similarly effective (Fig. 5a). As early as day 15 after DENV infection, a potent boost of NAb titers in both ZIKV-immune groups was developed.

However, elevated NAb titers were significantly higher in the ZIKVPF-10mo group compared with the ZIKVPR-2mo and naive groups at day 15 pi ($p = 0.0005$, $p < 0.0001$; two-way ANOVA Tukey test) and day 30 pi ($p = 0.0067$, $p = 0.0012$; two-way ANOVA Tukey test). As expected, this elevated ZIKV cross-reactive NAb levels decreased gradually overtime after 15 dpi in both ZIKV-immune groups. Nevertheless, the ZIKVPF-10mo group retained higher NAb titers until 90 dpi, while the titers of the ZIKVPR-2mo group returned to baseline levels. Of note, the NAb titers of the naive group were considered as negative in all timepoints and failed to neutralize ZIKV throughout DENV infection even at concentrated levels of the Abs (Fig. 5a). These results are confirmed by the behavior of neutralization kinetics by sigmoidal response curves where the ZIKVPF-10mo group retained elevated magnitude of ZIKV neutralization overtime (Supplementary Fig. 5).

To determine if the immune memory induced by different ZIKV strains play a role in the modulation of the cross-NAb response triggered by a subsequent DENV infection, NAb titers were measured against both ZIKV strains before and 30 days after DENV infection. The ZIKVPF-10mo group showed significant

higher NAb titers against both ZIKV strains compared with the ZIKVPR-2mo group before DENV infection ($p = 0.0093$, $p = 0.0141$; two-way ANOVA Tukey test) (Fig. 5b). Subsequently, DENV infection promote an equally eight-fold increase of NAb titers against both strains in the ZIKVPF-10mo group, significantly higher than the four-fold increase in the ZIKVPR-2mo group ($p = 0.0025$, $p = 0.0011$; two-way ANOVA Tukey test) (Fig. 5c). To rule out that difference in fitness between both ZIKV strains would bias the magnitude of the NAbs after DENV infection, we compared in parallel the NAb titers at 30 and 60 days after ZIKV infection (day 60 corresponds to the baseline of the ZIKVPR-2mo group). No significant differences were observed between ZIKV-immune groups in the NAb titers induced by both strains at the same timepoints after ZIKV infection (Supplementary Fig. 6). Altogether, these results demonstrate that DENV infection results in a significant increase in the magnitude and durability of the cross-NAb response against ZIKV in animals with a mid-convalescent period from ZIKV infection. The elicited changes in neutralization capacity were likely driven more by the longevity of the immune memory maturation and the associated memory recall of the ZIKV immunity than by a strict dependency of the specific pre-exposed ZIKV strain.

**Immune cell subsets modulated by ZIKV immunity**. We performed immunophenotyping by flow cytometry to assess the frequency, early activation, and proliferation of multiple immune cell subsets, and how these parameters are affected by the presence of pre-existing immunity to ZIKV on a subsequent DENV infection (Supplementary Figs. 7, 8, and 9 for gating strategy; Supplementary Table 2 for Ab panel). As part of the innate immune response, the frequency of dendritic cells (DCs) and natural killer (NK) cells subpopulations were measured. Plasmacytoid DCs (pDCs: Lin$^-$HLA-DR$^+$CD123$^+$) are known to respond to viral infection by production of IFN-α, while myeloid DCs (mDCs: Lin$^-$HLA-DR$^+$CD11c$^+$) interacts with T cells. The frequency of pDCs was not significantly altered by DENV infection in any group compared with baseline levels (Supplementary Fig. 10a). At day 2 pi, we detected a significant increase of mDCs in the ZIKVPF-10mo group ($p = 0.0082$; two-way ANOVA Dunnett test) (Supplementary Fig. 10b). Furthermore, we determined the frequency of NK subpopulations including: NKCD8, NKCD56, NKp30, and NKp46 (Supplementary Fig. 11). In general, no differences were detected between baseline and after DENV infection in all groups for all NK subpopulations and receptor expression with the exception of the ZIKVPR-2mo group that showed a significant increases in the following subpopulations: NKG2A$^+$NKp30 and NKp30$^+$NKp46$^+$ at 7 dpi ($p = 0.0495$, $p = 0.0006$; two-way ANOVA Dunnett test) and NKp46$^+$NKp30$^+$ at 7 and 10 dpi ($p = 0.0005$, $p = 0.0001$; two-way ANOVA Dunnett test) (Supplementary Fig. 11j, o, s).

We next investigated cell subsets that are part of the bi-phasic (humoral/cellular) adaptive immune response such as B (CD20 + CD3-) and T (CD3 + CD20-) cells. No differences were detected in the total B cells between groups following DENV infection compared with baseline levels (Supplementary Fig. 12a), but ZIKV-immune groups had elevated levels of activated B cells (CD20 + CD3-CD69 +) since baseline and a trend to increase these levels more than the naive group overtime (Supplementary Fig. 12b). We detected a significant decrease of proliferating B cells (CD20 + CD3-Ki67 +) in naive animals at 7 and 10 dpi ($p = 0.0031$, $p = 0.0345$; two-way ANOVA Dunnett test), while ZIKV-immune groups retained their proliferating levels (Supplementary Fig. 12c). Interestingly, the ZIKVPF-10mo group showed a significant increase of B cells that were proliferating and activated

simultaneously (CD20 + CD3-CD69 + Ki67 +) as early as in day 1 pi ($p = 0.0240$; two-way ANOVA Dunnett test) and maintained higher levels up to 10 dpi (Supplementary Fig. 12d). Together, these phenotyping results of B cells are consistent with the early and boosted production of binding and NAbs in the ZIKVPF-10mo group compared with naive animals. The frequency of the total T cells (CD3$^+$CD20$^-$) and CD4$^+$/CD8$^+$ T-cell subsets was comparable at all timepoints before and after DENV infection in all groups of animals (Supplementary Fig. 13a–c).

Previous studies have demonstrated that DENV- and ZIKV-specific CD4$^+$ and CD8$^+$ T cells are enriched in certain memory subsets[23,48]. Thus, we measured whether the early activation of T-cell subpopulations, such as effector memory (CD3$^+$CD4$^+$CD28$^-$CD95$^+$) and central memory (CD3$^+$CD4$^+$CD28$^+$CD95$^+$) T cells (T-EM and T-CM), within each T-cell compartment was modulated following DENV infection in presence or absence of convalescence to ZIKV (Fig. 6; Supplementary Fig. 7 for gating strategy). The ZIKVPF-10mo group showed a significant higher frequency of activated CD4$^+$ and CD8$^+$ T-EM (CD3$^+$CD4$^+$CD28$^-$CD95$^+$CD69$^+$ and CD3$^+$CD8$^+$CD28$^-$CD95$^+$CD69$^+$) following DENV infection compared with basal levels (CD4$^+$ T-EM at 7 and 10 dpi: $p = 0.0001$, $p = 0.0072$; CD8$^+$ T-EM at 2 and 7 dpi: $p = 0.0291$, $p = 0.0001$; two-way ANOVA Dunnett test) (Fig. 6a, d). Interestingly, the ZIKVPR-2mo group showed a very limited frequency and activation of the CD4$^+$ and CD8$^+$ T-EM compared with the ZIKVPF-10mo and naive groups. However, this group with an early convalescent period to ZIKV, contrary to the other two groups, showed a very limited but significant activation of CD8$^+$ T-CM (CD3$^+$CD8$^+$CD28$^+$CD95$^+$CD69$^+$) at day 7 and 10 pi ($p = 0.0007$, $p = 0.0147$; two-way ANOVA Dunnett test) (Fig. 6e). In contrast, naive animals did not show any significant activation of these memory cell subsets after DENV infection. Collectively, these results suggest that following DENV infection: (i) animals with a mid-convalescence ZIKV immunity have a more dynamic B-cell response, and are able to rapidly produce more activated effector memory T cells from both T-cell compartments; (ii) animals with an early convalescence to ZIKV induce activation of T-CM in the CD8$^+$ compartment with a very limited T-EM frequency and activation profile compatible with a contraction phase of the T cell compartments; (iii) and animals without previous exposure to ZIKV exhibit a limited B-cell response and minimal modulation of memory T cell subpopulations at early timepoints as the ZIKV-immune groups.

**T cell functional response is shaped by ZIKV immunity**. To further characterize the cross-reactive T cell response, we investigated if different convalescent periods of ZIKV immunity impacted the outcome of the effector role of CD4$^+$ and CD8$^+$ T cells following DENV infection. PBMCs were isolated and stimulated with peptide pools from DENV and ZIKV envelope (E) proteins and from ZIKV nonstructural proteins (ZIKV–NS) (Supplementary Table 4 for peptide sequences). Then, intracellular cytokine staining using flow cytometry analysis (Supplementary Fig. 14 for gating strategy; Supplementary Table 3 for Ab panel) was performed to quantify the production of effector immune markers, such as the cytotoxic marker CD107a, IFN-γ, and TNF-α by CD4$^+$ and CD8$^+$ T cells at baseline, 30, 60, and 90 days after DENV infection (Fig. 7).

To assess the ZIKV-primed specific- or cross-reactive effector T cell response, we studied the response against ZIKV or DENV stimuli before DENV infection. In general, before DENV infection, we found that the ZIKV-primed effector T cell response was higher in CD8$^+$ (Fig. 7m, q, u) than in CD4$^+$ (Fig. 7a, e, i) T cells. Of note, significant higher levels of CD107a, INF-γ, and

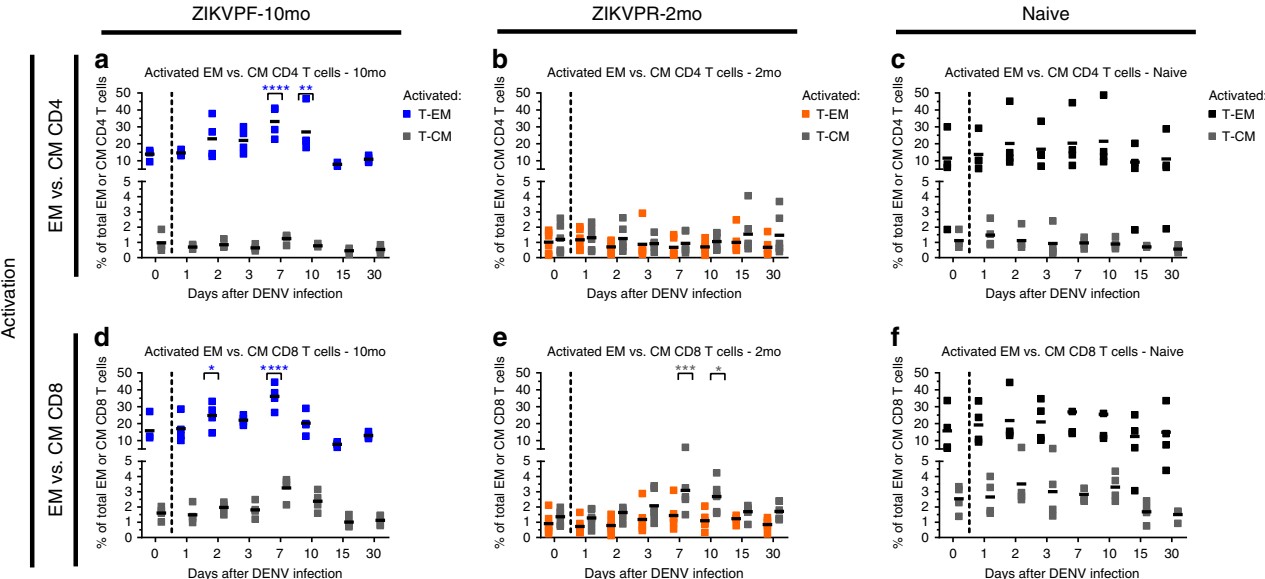

**Fig. 6** Activation of effector and central memory CD4+ and CD8+ T cells after DENV infection. Activation (CD69+) of effector memory (T-EM: CD3+ CD4+CD28−CD95+) and central memory (T-CM: CD3+CD4+CD28+CD95+) T cells within (**a–c**) CD4+ and (**d–f**) CD8+ T-cell compartments before and after DENV infection. Percent of cells were determined by immunophenotyping using flow cytometry (Supplementary Fig. 7 for gating strategy). Blue, orange, and black squares represent T-EM for ZIKVPF-10mo, ZIKVPR-2mo, and naive, respectively. Gray squares represent T-CM for each group. Short black lines mark mean value for each group per timepoint. Cutted line divide % of T-EM and T-CM cells quantified before and after DENV infection. Statistically significant differences within groups were determined using two-way ANOVA adjusted for Dunnett's multiple comparisons test (comparison of each group response at each timepoint versus baseline of the same group) including two families, and seven comparisons per family. Significant differences are reported as multiplicity adjusted p-values (*< 0.05, **< 0.01, ***< 0.001, ****< 0.0001). Asterisks represent significant difference between the corresponded timepoint and baseline within the same group. Source data are provided as a Source Data file

TNF-α-producing CD8+ T cells were found only in the ZIKVPF-10mo group before DENV infection (ZIKVPF-10mo vs ZIKVPR-2mo for CD107a: $p = 0.0002$; ZIKVPF-10mo vs naive for CD107a: $p = 0.0401$; ZIKVPF-10mo vs ZIKVPR-2mo for INF-γ: $p = 0.0020$; ZIKVPF-10mo vs ZIKVPR-2mo for TNF-α: $p = 0.0033$; ZIKVPF-10mo vs naive for TNF-α: $p = 0.0354$; two-way ANOVA Tukey test) (Fig. 7m, q, u). This basal effector response of CD8+ T cells in the ZIKVPF-10mo group is predominated by cross-reactive CD8+ T cells against DENV E protein. Very low effector T cell response against ZIKV NS proteins was detected for all groups (ZIKVPF-10mo > ZIKVPR-2mo > naive). In summary, results of T cell functional response before DENV infection suggest that a mid convalescence to ZIKV provoke a higher CD8+ T cell effector response capable to cross-react efficiently with DENV E protein.

After DENV infection, we were able to determine the modulation of the ZIKV-primed effector CD4+ and CD8+ T cell responses of ZIKV-immune groups and the de novo response of ZIKV-naive animals. The ZIKVPF-10mo and naive groups significantly boosted their CD107a expression in both T-cell compartments stimulated mainly by DENV E protein at 30 and up to 90 days after DENV infection (CD4+ T cells: ZIKVPF-10mo vs ZIKVPR-2mo: $p < 0.0001$ at 30 dpi, $p < 0.0001$ at 60 dpi; naive vs ZIKVPR-2mo: $p < 0.0001$ at 30 dpi, $p = 0.0018$ at 60 dpi; ZIKVPF-10mo vs Naïve: $p = 0.0204$ at 30 dpi. CD8+ T cells: ZIKVPF-10mo vs ZIKVPR-2mo: $p < 0.0001$ at 30 dpi, $p < 0.0001$ at 60 dpi, $p = 0.0008$ at 90 dpi; naive vs ZIKVPR-2mo: $p = 0.0039$ at 30 dpi, $p < 0.0001$ at 60 dpi; $p = 0.0081$ at 90 dpi; ZIKVPF-10mo vs naive: $p = 0.0194$ at 30 dpi; two-way ANOVA Tukey test) (Fig. 7b, c, n, o, p). Also, these groups boosted the CD107a cytotoxic signature reacting against ZIKV E and NS proteins by cross-reactive CD4+ T cells 30 days after DENV infection (ZIKVPF-10mo vs ZIKVPR-2mo: $p = 0.0025$ for ZIKV E, $p < 0.0001$ for ZIKV NS; Naïve vs

ZIKVPR-2mo: $p = 0.0025$ for ZIKV E, $p = 0.0002$ for ZIKV NS; two-way ANOVA Tukey test) (Fig. 7b).

The ZIKVPF-10mo group showed a remarkable significant increase of the IFN-γ-producing CD4+ T cells against DENV E protein since 60 dpi and is maintained up to 90 dpi compared with other groups (ZIKVPF-10mo vs ZIKVPR-2mo at 60 and 90 dpi: $p < 0.0001$, $p = 0.0024$; ZIKVPF-10mo vs Naïve at 60 and 90 dpi: $p < 0.0001$, $p = 0.0037$; two-way ANOVA Tukey test) (Fig. 7g, h), and was the only group with significant increase in the IFN-γ-producing CD8+ T-cell compartment at 60 dpi (ZIKVPF-10mo vs ZIKVPR-2mo: $p = 0.0253$; two-way ANOVA Tukey test) (Fig. 7s). On the other hand, the ZIKVPR-2mo group exhibited a significant increase of IFN-γ-producing CD4+ T cells earlier than other groups at 30 dpi (ZIKVPR-2mo vs ZIKVPF-10mo: $p < 0.0001$; ZIKVPR-2mo vs naive: two-way ANOVA Tukey test) (Fig. 7f). Interestingly, the naive group showed an increase of cross-reactive TNF-α-producing CD4+ T cells against ZIKV NS proteins 30 days after DENV infection (naive vs ZIKVPR-2mo: $p = 0.0359$; two-way ANOVA Tukey test) (Fig. 7j). The ZIKVPF-10mo group developed a significant effector T cell response by TNF-α-producing CD4+ T cells against DENV and ZIKV E proteins at 60 days after DENV infection (ZIKVPF-10mo vs ZIKVPR-2mo against DENV/ZIKV E protein: $p = 0.0163$, $p = 0.0172$; two-way ANOVA Tukey test) (Fig. 7k). Although all groups showed a boosted TNF-α effector response in the CD8+ T cell compartment up to 90 days after DENV infection, no significant differences between groups were observed.

Collectively, these results after DENV infection suggest that a mid convalescence to ZIKV translate in a more complete functional T cell response characterized by: (i) a cytotoxic CD107a+ phenotype directed to DENV E protein for both T-cell compartments comparable with the DENV-specific de novo response of the naive group, (ii) developed CD107a, IFN-γ, and

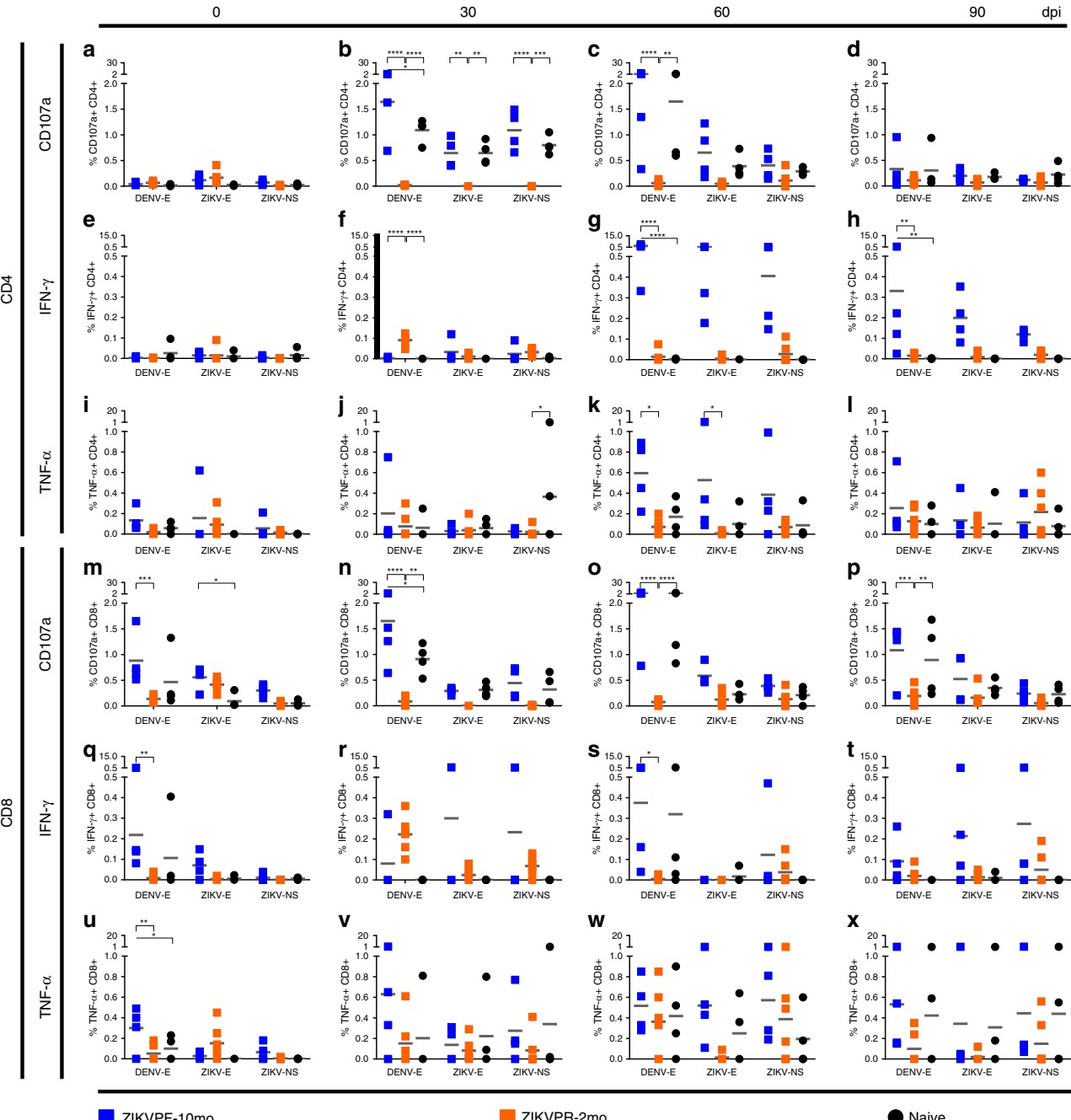

**Fig. 7** Longevity of ZIKV immunity shapes the T cell functional response. T cell functional effector response was determined by the quantification (%) of (**a–d; m–p**) CD107a-expressing and (**e–h; q–t**) IFN-γ or (**i–l; u–x**) TNF-α-producing CD4+ and CD8+ T cells before (0) and 30, 60, and 90 days after DENV infection. Responses to several peptide pools that encode for DENV and ZIKV envelope (E) proteins or ZIKV nonstructural (NS) protein were quantified. After antigenic stimulation, intracellular cytokine staining was performed using flow-cytometry analysis (Supplementary Fig. 14 for gating strategy). Individual symbols represent each animal per antigenic stimulation over time: blue squares (ZIKVPF-10mo), orange squares (ZIKVPR-2mo), and black circles (naive). Short gray lines mark mean value for each group. Statistically significant differences between groups were calculated using two-way ANOVA adjusted for Tukey's multiple comparisons test including three families, and three comparisons per family. Significant multiplicity adjusted *p*-values (*< 0.05, **< 0.01, ***< 0.001, ****< 0.0001) are shown. Asterisks represent significant difference between indicated groups. Source data are provided as a Source Data file

TNF-α-producing CD8+ T-cell effector response that cross-react efficiently with DENV E protein since baseline, and is boosted after DENV infection, (iii) and promoted the higher T cell effector response against ZIKV NS proteins. An early convalescence to ZIKV results in (iv) a very limited cytotoxic activity (limited expression of CD107a marker) which is in line with a very limited activation of the T-EM, and with failed capability to react efficiently against E or NS proteins. The ZIKV-naive group response was characterized by: (v) production of a DENV-specific de novo functional T cell response with similar magnitude between both T cell compartments, (vi) capable to cross-react against ZIKV E and NS proteins, (vii) and able to mount a DENV-specific cytotoxic CD107a+ phenotype.

## Discussion

We found that previous ZIKV infection modulates the immune response against subsequent DENV infection without an enhancement of DENV viremia nor pro-inflammatory status. This modulation is influenced by the longevity of ZIKV convalescence—more after longer ZIKV pre-exposure (Supplementary Discussion: Findings summary).

The aftermath of the recent ZIKV epidemic has been related to a remarkable decrease in DENV cases in Brazil[27], and also in most of Latin American and Caribbean countries (http://www.paho.org/data/index.php/es/temas/indicadores-dengue/dengue-nacional/9-dengue-pais-ano.html?start=2)[24]. To evaluate the hypothesis of a potential ZIKV–DENV cross-protection in humans will be necessary to characterize the immunological history of prospective cohorts[45], but human samples for this purpose are scarce yet. Because of this, NHPs are key to provide knowledge and anticipate different immunological scenarios when DENV epidemics re-emerge in human populations with previous immunity to ZIKV.

Animals with pre-existing ZIKV immunity do not show an enhancement of DENV-induced RNAemia. Previous ZIKV immunity is associated with a trend of less RNAemia days during subsequent DENV infection. This effect is more evident in animals with a ZIKV convalescence period of 10 months. Previous work reported that a period of early convalescence (56 days) to ZIKV (PRVABC59 strain) in rhesus macaques was associated with a significant increase of DENV-2 RNAemia at day 5 after DENV infection and a pro-inflammatory cytokine profile. However, very similar to our results, it was noteworthy a delay at early timepoints and an early clearance in late timepoints of the DENV-2 RNAemia in ZIKV-immune macaques in comparison to the naive ones[37]. Several factors may account for differences in the DENV RNAemia and pro-inflammatory status between early ZIKV convalescent cohorts in both studies, such as DENV strains, sample types, animals genetic heterogeneity, and cytokine panels (Supplementary Discussion: DENV RNAemia, Cytokine profile).

Due to limited availability of ZIKV-immune cohorts, we used animals infected with two different ZIKV strains for our subsequent challenge with DENV-2. However, extensive revision of the literature up-to-date reveals a broad consensus that these two contemporary ZIKV strains behave very similar from an antigenic point of view[11,49–51]. ZIKV strains were neutralized with same efficacy within each ZIKV-convalescent group, explained by the broadly neutralization activity against multiple ZIKV strains irrespective of the infecting strain[50]. However, the magnitude of the neutralization against both strains was statistically higher in animals exposed to DENV 10 months (mid convalescence) after ZIKV infection. These results suggest that the differences in the neutralization profile between the two ZIKV-immune groups are associated to the longevity of ZIKV convalescence, which may be attributable to the maturation of the cross-reactive immune memory elicited by the heterologous DENV infection and not to the antigenic differences or the different replication capabilities in rhesus macaques of those two pre-infecting ZIKV strains[16,52].

The period of convalescence further had an impact in the maintenance of the neutralization magnitude against ZIKV and DENV overtime. We observed a higher activation of the memory immune response characterized by transiently higher peak levels of serum NAbs against DENVs and ZIKV in ZIKVPF-10mo immune animals compared with ZIKVPR-2mo immune animals challenged with DENV-2. These results show the ability of DENV-2 to activate memory B cells (MBCs) stimulated by the previous ZIKV infection, but this activation is modest and short-lived compared with the robust and sustained activation of MBCs on secondary DENV infections[10,30,46,53]. On the other hand,

ZIKV NAbs decay overtime in ZIKV-immune animals after DENV infection, but animals with longer convalescence retain higher titers until the end of the study. Overall, these results demonstrate that pre-existing ZIKV immunity leads to a transient increase in NAb responses in animals challenged with DENV-2 compared with naive animals. This is in contrast with previous findings were ZIKV-convalescent macaques show a lack of an early and delayed anamnestic response overtime with limited induction of DENV NAbs compared with ZIKV-naive animals after DENV infection[54]. Interestingly, ZIKV-convalescent animals showed some degree of cross-neutralization against DENV-2 and DENV-4 before DENV infection, which is in line with previous findings in NHPs and humans (Supplementary Discussion: Neutralizing antibody response)[16,55].

T cells from DENV immunity are being implicated in mediating cross-protection against ZIKV[21–23], but their role and kinetics in the ZIKV–DENV scenario are poorly understood. We found that animals with a mid-convalescence to ZIKV developed an early activation of CD4$^+$ and CD8$^+$ effector memory T cells after DENV infection, similar to findings of DENV–ZIKV opposite scenario in humans[23]. On the other hand, the ZIKV early-convalescent group displays a modest activation (T-CM > T-EM) early after DENV infection associated with a possible contraction phase (Supplementary Discussion: T cells phenotyping)[56,57]. Moreover, we observed an IFN-γ phenotype of the functional response of CD8$^+$ T cells in animals with longer convalescence to ZIKV, which has been identified also in ZIKV-convalescent humans[58], and early after DENV challenge in DENV-vaccinated subjects protected from severe disease[59,60]. Strikingly, this response recognizes more efficiently peptides from DENV E protein than from ZIKV E protein (Supplementary Discussion: T cells functional response). In addition, an increased cytotoxic profile by CD107a-expressing CD4$^+$ and CD8$^+$ T cells in the ZIKV mid-convalescent group correlates with the early activation of CD4$^+$ and CD8$^+$ effector memory T cells and elevated levels of perforin release. Memory CD4$^+$ T cells are required to generate an effective humoral response against ZIKV[61]. Based on this, the higher proportion of DENV-E-reactive IFN-γ-producing CD4$^+$ T cells may play a role in the induction of the robust Ab response in the ZIKV mid-convalescent group against ZIKV and all DENV serotypes.

One limitation of our study is the utilization of low numbers of animals per group. However, fundamental and seminal contributions on ZIKV and ZIKV/DENV interactions have been obtained by using similar limited number of animals per group[16,17,62,63]. Another limitation is that our study monitored the immune response up to 90 days after DENV infection. Additional longitudinal studies are needed to test the immune response over longer periods of time, including subsequent DENV heterotypic challenges to evaluate the efficacy of the memory recall in cross-protection between serotypes. Finally, we cannot comment about the likelihood to increase or decrease susceptibility to develop DHF/DSS in the context of ZIKV immunity since DENV clinical manifestations in NHP models are limited and are characterized to be subclinical infections[28].

This study reinforces the usefulness of NHPs as a suitable model to characterize the immune response elicited by heterologous flavivirus infections. Our findings of highly cross-reactive response against DENV in presence of previous ZIKV immunity with no exacerbation of DENV pathogenesis may contribute to explain the decrease of detected DENV cases after ZIKV epidemic in the Americas. This scenario has been suggested recently using a fewer number of animals[38]. Furthermore, our data show a positive scenario that supports the implementation of ZIKV vaccine programs, since it suggests that a vaccine-acquired ZIKV-immunity may not worsen DENV pathogenesis, and may ameliorate immune response against a subsequent infection with DENV. Similarly, the

implementation of DENV vaccines is also supported in the context of previous ZIKV immunity, since ZIKV convalescence may boost the vaccine-acquired anamnestic immune response to DENV without predisposing to an enhanced pathogenesis. However, the selection of the vaccine schedule may be critical to induce the optimal immune response when more than one doses are planned.

## Methods

**Cell lines.** Aedes albopictus cells, clone C6/36 (ATCC CRL-1660), whole mosquito larva cells, were maintained in the Dulbecco Minimum Essential Medium (DMEM) (GIBCO, Life Technologies) supplemented with 10% fetal bovine serum (FBS) (Gibco) and 1% penicillin/streptomycin (P/S) (Gibco). C6/36 were used to produce previous ZIKV and DENV viral stocks with high titers in 150–175 cm$^2$ cell culture flasks (Eppendorf), and incubated at 33 °C and 5% $CO_2$. Vero cells, clone 81 (ATCC CCL-81), and African green monkey kidney epithelial cells were maintained in DMEM supplemented with 10% FBS and 1% of P/S, HEPES, L-glutamine, and nonessential amino acids (NEAA) in 75 cm$^2$ cell culture flasks, and incubated at 37 °C and 5% $CO_2$. Vero-81 cells were used for the cells monolayer in viral titrations by plaque assays and PRNT in flat bottom 24-well plates (Eppendorf).

**Viral stocks.** The DENV-2 New Guinea 44 (NGC) strain (kindly provided by Steve Whitehead, NIH/NIAID, Bethesda, MD, USA), known to replicate well in rhesus macaques, was used for the challenge in order to obtain comparative results with previous published studies from our group on DENV and ZIKV challenge studies[16,33]. We have standardized the assays to quantify this virus by Plaque assay, as described in our previous work[16]. The titer of DENV-2 for the challenge was 5 × 10$^7$ pfu/ml. In addition, ZIKV H/PF/2013 strain (kindly provided by CDC-Dengue Branch, San Juan, Puerto Rico), ZIKV PRVABC59 (ATCC VR-1843), DENV-1 Western Pacific 74, DENV-3 Sleman 73, and DENV-4 Dominique strains (kindly provided by Steve Whitehead from NIH/NIAID, Bethesda, MD, USA) were propagated in C6/36 cells, titrated, and used for PRNT assays.

**Viral titration plaque assay.** DENV titrations by plaque assay were performed seeding Vero-81 (~8.5 × 10$^4$ cells/well) in flat bottom 24-well cell culture well plates (Eppendorf) in supplemented DMEM the day before. Viral dilutions (tenfold) were made in diluent media [Opti-MEM (Invitrogen) with 2% FBS (Gibco) and 1% P/S (Gibco)]. Prior to inoculation, growth medium was removed and cells were inoculated with 100 μl/well of each dilution in triplicates. Plates were incubated for 1 h, 37 °C, 5% $CO_2$ and rocking. After incubation, virus dilutions were overlaid with 1 ml of Opti-MEM [1% carboxymethylcellulose (Sigma), 2% FBS, 1% of NEAA (Gibco), and P/S (Gibco)]. After 3–5 days of incubation (days vary between DENV serotypes), overlay was removed and cells were washed twice with phosphate buffered saline (PBS), fixed in 80% methanol (Sigma) in PBS, and incubated at room temperature (RT) for 15 min. Plates were blocked with 5% nonfat dry milk (Denia) in PBS for 10 min. Blocking buffer was discarded and 200 μl/well of primary antibodies mix [anti-E protein monoclonal antibody (mAb) 4G2 and anti-prM protein mAb 2H2 (kindly provided by Aravinda de Silva and Ralph Baric, University of North Carolina Chapel Hill, NC, USA), both diluted 1:250 in blocking buffer] were added and incubated for 1 h, 37 °C, 5% $CO_2$ and rocking. Plates were washed twice with PBS and incubated in same conditions with horseradish peroxidase (HRP)-conjugated goat anti-mouse secondary antibody (Sigma), diluted 1:1000 in blocking buffer. Plates were washed twice with PBS, and 150 μl/well of TrueBlue HRP substrate (KPL) were added, and plates were incubated from 1 to 10 min at RT until plaque-forming units (pfu) were produced and visible. Then 200 μl/well of distilled water were added to stop the substrate reaction, plates get dry, and pfu were counted to calculate viral titers.

**Macaques and viral challenge.** From 2008 to 2015, the Caribbean Primate Research Center (CPRC) funded a large DENV research program. Multiple studies made available several cohorts of rhesus macaques (Macaca mulatta) infected with different DENV serotypes in distinct timelines, and also naive cohorts were available as well. After our laboratories prioritized ZIKV research since 2016, DENV pre-exposed and naive cohorts were infected with ZIKV and pre-exposed animals became available for this study. All animals were housed within the Animal Resources Center facilities at the University of Puerto Rico-Medical Sciences Campus (UPR-MSC), San Juan, Puerto Rico. All the procedures were performed under the approval of the Institutional Animal Care and Use Committee (IACUC) of UPR-MSC and in a facility accredited by the Association for Assessment and Accreditation of Laboratory Animal Care (AAALAC file # 000593; Animal Welfare Assurance number A3421; protocol number, 7890116). Procedures involving animals were conducted in accordance with USDA Animal Welfare Regulations, the Guide for the Care and use of Laboratory Animals, and institutional policies to ensure minimal suffering of animals during procedures. All invasive procedures were conducted using anesthesia by intramuscular injection of ketamine at 10–20 mg kg$^{-1}$ of body weight. Rhesus macaques from the CPRC are very well genetically characterized from a common stock introduced in 1938 at Cayo Santiago, an islet located in the southeast of Puerto Rico. These macaques

with Indian genetic background are part of the purest colony used in the United States for comparative medicine and biomedical research[64].

The experimental design was based on 14 young adult male rhesus macaques divided into three cohorts. Cohort 1 (ZIKVPF-10mo): composed of four animals (5K6, CB52, 2K2, and 6N1) that were inoculated with 1 × 10$^6$ pfu/500 μl of the ZIKV H/PF/2013 subcutaneously[16] 10 months before DENV-2 challenge. Cohort 2 (ZIKVPR-2mo): composed of six animals (MA067, MA068, BZ34, MA141, MA143, and MA085) that were inoculated with 1 × 10$^6$ pfu/500 μl of the ZIKV PRVABC59 strain 2 months before DENV-2 challenge. Both ZIKV strains used for previous exposure of these groups are > 99.99% comparable in amino acid identity (Supplementary Table 1). Cohort 3 (naive): composed of four ZIKV/ DENV naive animals (MA123, MA023, MA029, and MA062) as a control group. Cohorts 1 and 3 were challenged on the same day, while cohort 2 was challenged 3 months later with the same stock of DENV-2. However, all samples were frozen and analyzed together, except for the immunophenotyping analysis.

The ages of all animals are within the age range for young adults rhesus macaques https://www.nc3rs.org.uk/macaques/macaques/life-history-and-diet/ (ZIKVPF-10mo: 6.8, 6.8, 5.8, and 5.9; ZIKVPR-2mo: 6.4, 6.5, 5.2, 4.3, 5.6, and 5.5; naive: 4.8, 6.6, 6.8, and 5.7). Prior to DENV-2 challenge, all animals were subjected to quarantine period. All cohorts were bled for baseline and challenged subcutaneously (deltoid area) with 5 × 10$^5$ pfu/500 μl of DENV-2 New Guinea 44 strain. After DENV-2 challenge, all animals were extensively monitored by trained and certified veterinary staff for evidence of disease and clinical status: external temperature (°C) with an infrared device (EXTECH Instruments, Waltham, MA, USA), weight (Kg), CBC, and CMP. All animals were bled once daily from day 1 to day 10 and after that on days 15, 30, 60, and 90 dpi. In all timepoints, the blood samples were used for serum separation (baseline, 7, 30, 60, 90 dpi only). PBMCs were collected at same timepoints using CPT tubes (BD-Biosciences, San Jose, CA, USA) containing citrate. Additional heparin samples were obtained for immunophenotyping by flow cytometry using fresh whole blood. Figure 1 shows the experimental design and samples collection timeline.

**DENV RNAemia.** DENV viral RNA extraction was performed from acute serum samples (baseline, 1–10, and 15 dpi) using a QIAamp Viral RNA mini kit (Qiagen Inc, Valencia, CA, USA) according to the manufacturer's instructions. RNAemia levels were measured by a One-Step qRT-PCR detection kit (Oasig, Primerdesign Ltd., UK), and using a DENV RT primer/probe Mix kit (Genesig, Primerdesign Ltd., UK) according to the manufacturer's protocol (catalog no. oasig-onestep). Primers are designed to target the 3′ untranslated region (3′ UTR) of all four DENV serotypes, and have 100% homology with over 95% of reference sequences contained in the NCBI database. Assays were performed in an iCycler IQ5 Real-Time Detection System with Optical System Software version 2.1 (Bio-Rad, Hercules, CA, USA). Limit of detection (LOD) was 20 copies per milliliter. Furthermore, in order to correlate RNAemia levels with DENV pathogenesis, we monitored the clinical status for injury and/or clinical manifestations. Complete blood counts (CBC) were performed for all animals in several timepoints (baseline, 7, and 15 dpi) to determine the absolute number (106 cells/ml) and percent (%) of lymphocytes (LYM), monocytes (MON), white blood cells (WBC), neutrophils (NEU), and platelets (PLT). Also, comprehensive metabolic panel (CMP) was evaluated in several timepoints (baseline, 7, 15, and 30 dpi) to measure concentration (U/L) of alkaline phosphatase and liver enzymes alanine aminotransferase (ALT) and aspartate aminotransferase (AST).

**ELISA.** Seroreactivity to DENV and cross-reactivity to ZIKV was measured at different timepoints before and after DENV-2 challenge. DENV–IgM (Focus Diagnostics, Cypress, CA, USA) was quantified at baseline, 5, 10, 15, and 30 dpi. DENV–IgG was quantified at baseline, 7, 15, 30, 60, and 90 dpi (Focus Diagnostics, Cypress, CA, USA). To determine the modulation of serological profile against ZIKV, we assessed: levels of anti-ZIKV IgM (InBios, Seattle, WA, USA) at baseline, 5, 10, 15 and 30 dpi; anti-ZIKV IgG (XPressBio, Frederick, MD, USA) at baseline, 7, 15, 30, 60, and 90 dpi; anti-ZIKV NS1-IgG (Alpha Diagnostics, San Antonio, TX, USA) at baseline, 30, 60, and 90 dpi (including additional timepoints prior baseline for both ZIKV-immune groups); and anti-ZIKV EDIII-IgG (Alpha Diagnostics International, San Antonio, TX, USA). All ELISA-based assays were performed following the manufacturers' instructions. This serological characterization allows us to assess the dynamics of DENV and ZIKV cross-reactivity, but without discerning between cross-reactive binding Abs and cross- or type-specific neutralizing Abs.

**Plaque reduction neutralization test.** Selected serum samples (baseline, 30, 60, and 90 dpi) were challenged to neutralized ZIKV (H/PF/2013, PRVABC59), DENV-1 Western Pacific 74, DENV-2 NGC 44, DENV-3 Sleman 73, and DENV-4 Dominique strains. For the infecting serotype (DENV-2) and ZIKV, the NAbs were measured in early timepoints as well (7 and 15 dpi). For the PRNT, serum samples were inactivated, diluted (twofold), mixed with a constant inoculum of virus (volume necessary to produce ~ 35 pfu/well) and then incubated for 1 h at 37 °C and 5% $CO_2$. After incubation, virus–serum mix dilutions were added to Vero-81 cells monolayer in flat bottom 24-well plates seeded the day before for 1 h at 37 °C and 5% $CO_2$, finally overlay medium was added and incubated by several days (serotype dependent). The results were reported as PRNT60 titers, NAb titer

capable of reduce 60% or more of DENV serotypes or ZIKV strains pfu compared with the mock (control of virus without serum). A PRNT60 1:20 titer was considered a positive Neut titer, and < 1:20 as a negative Neut titer. Non-neutralizing titers < 1:20) were assigned with one-half of the limit of detection for graphs visualization.

**Multiplex cytokine profile**. A total of eight cytokines/chemokines were measured (pg ml$^{-1}$) by Luminex at baseline, 1, 2, 3, 5, 10, 15, and 30 dpi, including: interferon alpha (IFN-α), interleukin-6 (IL-6), monokine induced by IFN-gamma (MIG/CXCL9), monocyte chemoattractant protein 1 (MCP-1/CCL2), macrophage inflammatory protein 1-beta (MIP-1β/CCL4), IL-1 receptor antagonist (IL-1RA), C–X–C motif chemokine 10 (CXCL10/IP-10), and perforin. The multiplex assay was conducted as previously described[16,65].

**Immunophenotyping**. Flow cytometry (MACSQuant Analyzer 10, Miltenyi Biotec) analysis was performed to determine the frequency, activation, and proliferation of cell populations of the innate and adaptive immune response based on the phenotyping strategy of a previous study[16] (Supplementary Figs. 7, 8, and 9 for gating strategy; Supplementary Table 2 for Ab panel). Phenotypic characterization of macaque PBMCs from fresh whole blood samples was performed by 8-multicolor flow cytometry using fluorochrome-conjugated Abs at several timepoints (baseline, 1, 2, 3, 7, 10 dpi; and 15 and 30 dpi for B-/T-cell panel only). Single cells (singlets) were selected by their FSC area (FSC-A) and height (FSC-H) patterns. Lymphocytes (LYM) were gated based on their characteristic forward and side scatter pattern (FSC, SSC). T cells were selected gating on the CD3$^+$ population. CD4$^+$ and CD8$^+$ T cells were defined as CD3$^+$CD4$^+$ and CD3$^+$CD8$^+$, respectively. Naive (N; CD28$^+$CD95$^-$), effector memory (EM; CD28$^-$CD95$^+$), and central memory (CM; CD28$^+$CD95$^+$) T-cell subpopulations were determined within CD4$^+$ and CD8$^+$ T cells. B cells were defined as CD20$^+$CD3$^-$. The activation of B- and T-cell memory subpopulations (EM and CM) was assessed by the presence of the early activation marker CD69. Proliferation of the total and activated B cells was quantified by the expression of the intracellular marker Ki67. Natural killer (NK) cells were defined as CD3$^-$CD20$^-$CD14$^-$, and analyzed by the double-positive expression of the following NK cell markers: CD8, CD56, NKG2A, NKp30, and NKp46 (Supplementary Fig. 9 for gating strategy). Dendritic cells (DC) were separated in two populations within the Lineage-DR + (HLA-DR$^+$ CD3$^-$ CD14$^-$ CD16$^-$ CD20$^-$ CD8$^-$ NKG2A$^-$) by the expression of CD123 (plasmacytoid, pDC) or CD11c (myeloid, mDCs) (Supplementary Fig. 8 for gating strategy). Then, DC percentages were calculated from the total PBMCs (total events of the DC subpopulation divided by total PBMCs and multiplied by 100). The phenotyping assays were optimized and performed as previously published[16,33,66].

**T-cell functional response**. Intracellular cytokine staining of macaques PBMCs was performed by multicolor flow cytometry using methods previously described (Supplementary Fig. 14 for gating strategy; Supplementary Table 3 for Ab panel)[16,66]. Functional effector response of CD4$^+$ and CD8$^+$ T cells was measured before and after DENV infection. Antigen-specific CD4$^+$ and CD8$^+$ T-cell effector responses were measured at baseline to determine basal levels in presence (ZIKVPF-10mo, ZIKVPR-2mo) or absence (naive) of previous immunity to ZIKV. Also, 30, 60, and 90 dpi were assessed to determine how this pre-existing functional response is modulated after DENV infection and if is maintained over time. For peptide pools stimulation, PBMCs were stimulated for 6 h at 37 °C and 5% CO$_2$. The peptides used for DENV-E, ZIKV-E, and ZIKV–NS were 15-mers overlapped by 10 amino acids at 1.25 µg/ml, 2.5 µg/ml, 475 ng/ml per peptide, respectively (Supplementary Table 4 for peptide sequences). The stimulation with peptides was performed in presence of brefeldin A at 10 µg/ml. After the cells were stained for the following markers: CD3, CD4, CD8, CD20 (excluded), CD107a (functional cytotoxicity). Levels of IFN-γ and TNF-α also were measured in gated lymphocytes cell populations. Samples were measured, and data were collected on a LSRII (BD).

**Statistical analysis**. Statistical analyses were performed using GraphPad Prism 7.0 software (GraphPad Software, San Diego, CA, USA). The statistical significance between the means of all groups were determined using two-way ANOVA multiple comparison Tukey test, and to compare each mean against the baseline mean within same group two-way ANOVA multiple comparison Dunnett test was performed. The total number of families and comparisons per family used for adjustments are depicted in each figure legend. Significant multiplicity adjusted $p$-values (*< 0.05, **< 0.01, ***< 0.001, ****< 0.0001) show statistically significant difference between groups (Tukey test) or timepoints within a group (Dunnett test).

**Reporting summary**. Further information on research design is available in the Nature Research Reporting Summary linked to this article.

## Data availability

All relevant data are in main Figures and Supplementary Information, any additional details are available from the authors upon request. The raw data from all main and supplementary figs. are provided as a Source Data File.

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

## Acknowledgements

We thank all the staff of the Caribbean Primate Research Center and Animal Resources Center for their continuous support with the sample collection, schedule, and monitoring of the animals. Authors recognize the support provided by Dr. Elmer Rodriguez reviewing the statistics. This work was supported by the following grants: 2 P40 OD012217 and 2U42OD021458-15 to C.A.S. and M.I.M., K22AI104794 to J.D.B., P51OD011133 (L.G.), HHSN272201400045C to D.W., and R25GM061838 to E.X.P.-G.

## Author contributions

C.A.S. and E.X.P.-G. developed the experimental design. I.V.R. and M.I.M. supervised and performed sample collection and animals monitoring. E.X.P.-G., P.P., C.S.-C., M.A.H., A.O.-R., V.H., L.P., L.C., and T.A. performed the experiments. E.X.P.-G., C.A.S., V.H., M.A.H., I.V.R., M.I.M., L.J.W., A.d.S., and D.W. analyzed the data. E.X.P.-G. and C.A.S. drafted the paper. C.A.S., E.X.P.-G., D.W., A.K.P., J.D.B., M.A.H., L.G., L.J.W., and A.d.S. revised the paper.

## Additional information

**Competing interests:** The authors declare no competing interests.

