## [Peer Review File · Nature Communications]

Editorial Note: A figure on page 18 in this Peer Review File has been amended to remove third-party material where no permission to publish could be obtained.

Reviewers' Comments:

Reviewer #1:

Remarks to the Author:

In this manuscript, Perez-Guzman et al. describe a nonhuman primate (NHP) study that investigates the effect of a prior ZIKV infection on the outcome of subsequent DENV infection. By using 2 cohorts of animals, infected with ZIKV either 2 or 10 months prior to DENV challenge, they investigate the effect of the time window on the outcome of DENV infection. The investigators demonstrate that animals previously infected with ZIKV don't have evidence an increased DENV disease course; in contrast, the available data suggest that DENV infection was slightly attenuated, which is an important finding. To get to the mechanism of this, the investigators have performed very detailed immunological analyses, and demonstrate that ZIKV infected animals, relative to naïve animals that are exposed to DENV, have improved humoral and cellular immune responses against DENV. Importantly, this effect of enhanced anti-DENV immune responses was more pronounced in the 10 month than the 2-month group, which indicates maturation and improvement of the immune responses occurred over time. As the authors pointed out, further studies with even longer intervals between ZIKV and DENV are warranted, but for now, the findings in this manuscript are already important, are very timely, and have high merit.

The authors nicely link observations of this NHP study to results from human studies that already indicated that prior ZIKV infection may attenuate DENV infection; but while human observational studies have limitations to conclusively demonstrate cause-and-effect, the NHP study was designed to directly test this and investigate possible mechanisms. The authors have to be applauded for the very extensive analyses of cross-reactive immune responses, including cell-mediated immune responses, which other research groups have often not looked at. They have also performed detailed and appropriate statistical analyses.

Overall, this is an excellent study, and of high importance for the ZIKV/DENV research field, and that merits publication in Nature Communications.

I only have a few minor comments for improvement:

- While the authors do an excellent job in describing all the findings so that the final result is convincing, the manuscript's main text is very long and thus upon publication, may distract some readers. The manuscript could benefit from condensing it. For example, some details are explained or provided both in the main text as well as in the figure legend and/or materials/methods, and thus can often be shortened in the main text, without having to lose any of the details for those readers who want to see it.

- Figure 1, and also in the materials & methods: reading it, I get the impression that the same blood tube was used for both serum and PBMC isolation. PBMC isolation assumes there was an anticoagulant, in which case plasma (instead of serum) was collected. Or did the investigators collect separate blood tubes for serum (which by definition has no anti-coagulant) versus PBMC (with anti-coagulant; if so, which anticoagulant)?

- Fig 2a, I have the impression that the mean is calculated from the original (untransformed) values? For viral load data, it makes sense (and it's commonly done) to do a log-transformation and then calculate the means, as it would reduce the impact of outliers, and give the data more a Gaussian distribution. I am wondering if the mean of logtransformed values would show a difference between the groups so I recommend the authors to explore this; it's possible that the calculation of the AUC of the log-transformed RNA levels may show a difference, especially considering that the 10-month group has animals with shortened viremia. Such graph with AUC values, if it shows differences, could

potentially be added to figure 2.

- Please mention in the methods, and indicate in figure 2, the LOD for the RT-PCR assay.

Reviewer #2:

Remarks to the Author:

This is a timely, interesting and comprehensive study of the role prior immunity to ZIKV plays in heterologous protective immunity against DENV infection demonstrating the potential role for both B cell and T cell crossreactivity in a macaque model. It shows novel data suggesting that this cross protection takes time to mature following ZIKV infection. These findings may help to explain recent epidemiological studies from some South American countries that indicate a reduction of DENV cases during and after ZIKV epidemics. The results presented here provide strong evidence that both B cell and T cell crossreactivity exists between ZIKV and DENV leading to largely beneficial immune modulation decreasing strong innate responses. Leading to early activation of DENV-specific Nab and activated T cell responses in particular perforin upregulation which could be important in clearance of DENV. There was very little evidence of a harmful heterologous immune effect as has been shown between different Dengue serotypes (only a slight increase in ALT.) Here, the authors show that polyfunctional effector response to DENV is shaped by the longevity of ZIKV-immunity and suggested that the presence of cross-reactive T cells between these two heterologous infections may provide cross-protective effect. These are very important studies that set the stage for detailed analysis in humans of which crossreactive B cell and CD4 and CD8 T cell epitopes may mediate this crossreactive protective effect so that they be included in ZIKV vaccines.

Minor;

There are some typographical errors that should be corrected throughout the manuscript

Reviewer #3:

Remarks to the Author:

In this manuscript, Perez-Guzman et al report on the clinical, virological, and immunological observations after challenge with DENV2 strain NGC in cohorts of rhesus macaques that were either ZIKV-naïve or previously challenged with ZIKV strains 2 or 10 mo previously. The authors compared serum viral RNA levels, frequencies of various PBMC subpopulations in peripheral blood, and antibody and T cell responses to DENV and ZIKV. The authors concluded that prior ZIKV immunity did not promote an increase in DENV viremia but improved the antibody and cell-mediated immune responses to DENV with greater efficiency after a longer interval.

Major comments:

1. Improvement of immune responses is not clearly defined in the manuscript, and, in fact, there is substantial evidence that the magnitude of the immune response to flaviviruses is an unreliable marker of protective immunity. This term should be avoided (e.g., in the title). Similarly, rather than concluding that their data provide evidence of a non-detrimental effect (lines 48-49), it would be more accurate given the very small sample size to state that they found no evidence of a detrimental effect.
2. The execution and presentation of the statistical analyses warrant substantial revision. The approach used for comparison of multiple groups (ANOVA with Tukey test) is reasonable, however,

this is appropriate only if the data are normally distributed and appropriate transformations have been applied to the data. In several cases, e.g., serum viral RNA levels and neutralizing antibody titers, log transformation did not appear to be used for calculation of means and statistical comparisons. P values are provided throughout the text for the post-hoc pairwise comparisons to 4 significant digits, and the authors suggest that smaller p values indicate more biologically significant findings. This needs to be reconsidered, particularly in light of the small sample sizes (4-6 per group). Finally, the authors should review all figures to ensure that individual data points or an estimate of the variance has been provided wherever appropriate.

3. The authors have not addressed the limitations of their study; this should be included in the Discussion. Overall, the Discussion is excessively long, repeats background information addressed in the Introduction, and is overly speculative given the limited differences observed in the study.

4. The comparison of 2 mo and 10 mo intervals between ZIKV and DENV challenge is a major focus of the authors' conclusions. While the description of the findings in these two groups is accurate, the authors do not adequately address differences between these groups at baseline (time of DENV challenge) and alternative explanations for their findings. Some additional methodological details are essential. Did the two different ZIKV challenge strains perform similarly- e.g., duration and magnitude of viremia, magnitude of antibody titers at equivalent time points, etc.? This is particularly relevant since figure 5 shows higher neutralizing antibody titers to ZIKV at 10 mo post-infection in the ZIKVVPF-10mo group than at 2 mo post-infection in the ZIKVPR-2mo group and figure S2 shows higher anti-ZIKV NS1 IgG; differences in the initial immune response to ZIKV (despite equivalent challenge doses as PFU) would potentially explain the differences observed after DENV challenge. Were all three cohorts challenged with DENV at the same time, so that assays on fresh PBMC were done concurrently? (This latter point is critical to the interpretation of some differences in flow cytometry data at baseline, such as CD69 staining shown in figure 6, that are otherwise difficult to understand.) How were groups "matched" for age and sex of animals, especially when the sample sizes were not identical? If groups were matched at the time of DENV challenge, could the 8 mo difference in ages at the time of ZIKV challenge have contributed to differences between the groups? (The ages and sexes of each animal at the time of DENV challenge could be included in the supplementary material.)

5. The authors found no significant differences in viremia between groups. This supports their conclusion that they did not observe an enhancement of viremia, and their discussion of potential explanations for the difference from published data is adequate. It is also fair to point out trends toward lower viremia at days 1 and 8, given the low statistical power of the study, but the authors overinterpret these data and speculate excessively in the Discussion about potential mechanisms for a non-significant difference. Data are also unnecessarily duplicated between figure 2 and table S2. Figure S4 re-analyzes the viremia and neutralizing antibody data; although the authors present this analysis (lines 313-334) as identifying an inverse correlation between these parameters, the analysis is inadequate to support this conclusion- mean values are shown for each group without a correlation analysis at the level of individual animals, and no statistical tests are applied to the data. Figure S4 should be removed or a more detailed analysis should be conducted to address the authors' hypothesis.

6. The authors found higher IFN-alpha and IL-6 levels in the ZIKV-naïve group and higher perforin and CXCL10 levels in the ZIKVVPF-10mo group. The overall description of these data, however, is somewhat inaccurate. IFN-alpha would not usually be included among the pro-inflammatory cytokines, for example. More significantly, the authors did not analyze levels of those cytokines reported by George et al to be increased in ZIKV-immune animals, IL-8 and IP-10, and as a result cannot compare results. This should be noted in the Discussion.

Minor comments:

7. Line 70- The statement that DENV possesses one of the highest mortality rates among arboviruses is inaccurate.
8. Figure S2- The time points after ZIKV infection in panel e should be explained in the legend.
9. Line 294- Although neutralizing antibody titers remained higher in the ZIKVPR-10mo group, the rate of decay appears to be identical to the ZIKVPR-2mo group.
10. Line 338- The meaning of heterologous ZIKV strains in ZIKV-immune groups is unclear. Also, the authors cannot draw conclusions about long-lasting antibody responses in the ZIKVPR-2mo group.
11. Figure 5b- The authors do not provide sufficient context for why they analyzed EC50 levels. In addition, it is unclear if/how this differs from PRNT50 calculations, and the symbols in the chart are not defined in the legend.
12. Figure S5- Neutralization curves for days 15-60 likely do not allow reliable modeling of PRNT60.
13. Representative flow cytometry and gating strategies should be provided for the analyses of DC and NK cells.
14. Figure S8 and line 391- NK8 and NK56 should be CD8 and CD56.
15. Lines 401 and 410- The statistical analyses performed did not test for differences between groups.
16. Line 411-413- The statement regarding B cell phenotyping data is accurate for the ZIKVPR-10mo group but not for the ZIKVPR-2mo group.
17. Figure S6- The gating of EM, CM, and naïve subpopulations is incorrectly labeled.
18. Figure S10- The percentage of CD69+ cells in total T cells does not appear to be consistent with the results in CD4 and CD8 subsets (e.g., ZIKVPR-2mo group). The order of symbols differs in panel d- is this correct?
19. The authors tested for T cell responses to ZIKV NS1 peptides and not to other non-structural proteins. This should be noted clearly in the text.
20. All plots of antigen-specific T cell responses show the frequencies of cells expressing individual effector functions; the authors do not present data on polyfunctional responses, i.e., expression of multiple effector functions by individual T cells. The text should be clarified or the data on such responses should be included, at least in summary form.
21. Lines 556-557- There is inadequate data to support this statement, and certainly not to draw a strong conclusion.
22. Lines 575-576- The data presented in figure 5c do not support this statement.
23. Line 839- Animals were (likely) not bled continuously.

24. Line 947- The authors should clarify the number of comparisons used for adjustment.

Point by Point answers to the Reviewers' comments (*in Italics*)

• Reviewer #1 (Remarks to the Author):

In this manuscript, Perez-Guzman et al. describe a nonhuman primate (NHP) study that investigates the effect of a prior ZIKV infection on the outcome of subsequent DENV infection. By using 2 cohorts of animals, infected with ZIKV either 2 or 10 months prior to DENV challenge, they investigate the effect of the time window on the outcome of DENV infection. The investigators demonstrate that animals previously infected with ZIKV don't have evidence an increased DENV disease course; in contrast, the available data suggest that DENV infection was slightly attenuated, which is an important finding. To get to the mechanism of this, the investigators have performed very detailed immunological analyses, and demonstrate that ZIKV infected animals, relative to naïve animals that are exposed to DENV, have improved humoral and cellular immune responses against DENV. Importantly, this effect of enhanced anti-DENV immune responses was more pronounced in the 10 month than the 2-month group, which indicates maturation and improvement of the immune responses occurred over time. As the authors pointed out, further studies with even longer intervals between ZIKV and DENV are warranted, but for now, the findings in this manuscript are already important, are very timely, and have high merit.

The authors nicely link observations of this NHP study to results from human studies that already indicated that prior ZIKV infection may attenuate DENV infection; but while human observational studies have limitations to conclusively demonstrate cause-and-effect, the NHP study was designed to directly test this and investigate possible mechanisms. The authors have to be applauded for the very extensive analyses of cross-reactive immune responses, including cell-mediated immune responses, which other research groups have often not looked at. They have also performed detailed and appropriate statistical analyses.

Overall, this is an excellent study, and of high importance for the ZIKV/DENV research field, and that merits publication in Nature Communications.

Authors appreciate these positive and stimulating comments. Authors are happy to see that the reviewer shares the same level of enthusiasm about this manuscript. Authors completely agree that not only are important findings, including extensive data analysis from most important immune response branches, but also the timing in which this may be available to the field will expand the understanding of current epidemiological settings after ZIKV epidemic in the Americas before DENV epidemics raise again. As highlighted by the reviewer, this work in NHPs may anticipate relevant observations that will take longer or not possible to elucidate from human cohorts.

I only have a few minor comments for improvement:

- While the authors do an excellent job in describing all the findings so that the final result is convincing, the manuscript's main text is very long and thus upon publication, may distract some readers. The manuscript could benefit from condensing it. For example, some details are explained or provided both in the main text as well as in the figure legend and/or materials/methods, and thus can often be shortened in the main text, without having to lose any of the details for those readers who want to see it.

Authors are agree with this comment. The new version has been shortened keeping the fundamental findings without losing needed details. Authors look forward to seeing the reviewer's comments on the new version.

- Figure 1, and also in the materials & methods: reading it, I get the impression that the same blood tube was used for both serum and PBMC isolation. PBMC isolation assumes there was an anticoagulant, in which case plasma (instead of serum) was collected. Or did the investigators collect separate blood tubes for serum (which by definition has no anti-coagulant) versus PBMC (with anti-coagulant; if so, which anticoagulant)?

Authors are agree with the reviewer that in the way the information was provided was not clear enough. This has been corrected in the method section that the PBMCs were collected using tubes with citrate as anticoagulant and serum was collected with different tubes. (Lines 1236-1237 in track changes version)

- Fig 2a, I have the impression that the mean is calculated from the original (untransformed) values? For viral load data, it makes sense (and it's commonly done) to do a log-transformation and then calculate the means, as it would reduce the impact of outliers, and give the data more a Gaussian distribution. I am wondering if the mean of log transformed values would show a difference between the groups so I recommend the authors to explore this; it's possible that the calculation of the AUC of the log-transformed RNA levels may show a difference, especially considering that the 10-month group has animals with shortened viremia. Such graph with AUC values, if it shows differences, could potentially be added to figure 2.

The reviewer is correct, the viremia curve was generated using the original untransformed value. In the reviewed version the viremia graph type was changed and the transformed viremia values were used (see below). Now the individual animals per group are better displayed. In addition, the AUC was calculated using the log-transformed RNA levels. The statistical analysis was repeated, and although the data shows a more Gaussian distribution, no statistically significance differences were observed in the viremia peak among groups. As the reviewer anticipated, the AUC graph is showing clearer the effect of the previous ZIKV immunity in DENV viremia. The AUC trend to be lower in both ZIKV-immune groups. In addition, the AUC figure shows a delay in the peak viremia set point in both ZIKV-immune groups (switch from day 2 to days 5 and 6) followed by higher, but non-significant viremia levels compared to the naïve group and by an early viremia clearance in both ZIKV-immune groups. After considering the reviewer suggestion the AUC figure is provided as Supplementary Figure 2. Authors would like to know if the reviewer is in agreement to add the AUC figure.

DENV RNAemia

DENV-2 RNAemia - Area Under the Curve

- Please mention in the methods, and indicate in figure 2, the LOD for the RT-PCR assay.

The LOD is 20 copies per ml. This has been added to the RNAemia figure 2 and methods section. (Lines 1247 in track changes version)

Reviewer #2 (Remarks to the Author):

This is a timely, interesting and comprehensive study of the role prior immunity to ZIKV plays in heterologous protective immunity against DENV infection demonstrating the potential role for both B cell and T cell crossreactivity in a macaque model. It shows novel data suggesting that this cross protection takes time to mature following ZIKV infection. These findings may help to explain recent epidemiological studies from some South American countries that indicate a reduction of DENV cases during and after ZIKV epidemics. The results presented here provide strong evidence that

both B cell and T cell crossreactivity exists between ZIKV and DENV leading to largely beneficial immune modulation decreasing strong innate responses. Leading to early activation of DENV-specific Nab and activated T cell responses in particular perforin upregulation which could be important in clearance of DENV. There was very little evidence of a harmful heterologous immune effect as has been shown between different Dengue serotypes (only a slight increase in ALT.) Here, the authors show that polyfunctional effector response to DENV is shaped by the longevity of ZIKV-immunity and suggested that the presence of cross-reactive T cells between these two heterologous infections may provide cross-protective effect. These are very important studies that set the stage for detailed analysis in humans of which crossreactive B cell and CD4 and CD8 T cell epitopes may mediate this crossreactive protective effect so that they be included in ZIKV vaccines.

Authors acknowledge the positive comments of this reviewer emphasizing the timely and detailed work presented in this manuscript that provides insights of B and T cells cross-reactivity. More importantly, the reviewer highlights the implication of these findings for the development of effective vaccines that ideally will promote cross-reactive protection between DENV and ZIKV. Particularly, the use of both ZIKV strains in this study became an important strength to support current ZIKV vaccine candidates, since the majority of them in clinical trials are based in the H/PF/2013 and PRVABC59 ZIKV strains used in this study. Moreover, these findings suggest that the schedule between multiple doses of a vaccine or time elapsed between vaccination and infection with the wild-type ZIKV or DENV will modulate the type-specific or cross-reactive vaccine-acquired anamnestic immune response.

Minor;

There are some typographical errors that should be corrected throughout the manuscript

Thank you to point this, so the manuscript was extensively reviewed to correct typographical errors.

Reviewer #3 (Remarks to the Author):

In this manuscript, Perez-Guzman et al report on the clinical, virological, and immunological observations after challenge with DENV2 strain NGC in cohorts of rhesus macaques that were either ZIKV-naïve or previously challenged with ZIKV strains 2 or 10 mo previously. The authors compared serum viral RNA levels, frequencies of various PBMC subpopulations in peripheral blood, and antibody and T cell responses to DENV and ZIKV. The authors concluded that prior ZIKV immunity did not promote an increase in DENV viremia but improved the antibody and cell-mediated immune responses to DENV with greater efficiency after a longer interval.

Major comments:

1. Improvement of immune responses is not clearly defined in the manuscript, and, in fact, there is substantial evidence that the magnitude of the immune response to flaviviruses is an unreliable marker of protective immunity. This term should be avoided (e.g., in the title). Similarly, rather than concluding that their data provide evidence of a non-detrimental effect (lines 48-49), it would be more accurate given the very small sample size to state that they found no evidence of a detrimental effect.

Authors used the word “improvement” in the sense of a change that makes something better or more valuable. The authors assumed the term “improvement” was defined in the abstract (Lines 42-45 original version).

“We found that previous ZIKV exposure improves the antibody and cell mediated immune responses against DENV and that the time interval between infections impacted the magnitude and durability—more efficient after longer ZIKV pre-exposure—of the immune response”

Additionally, we provided a summary supporting the term “improvement” (Lines 729-735, original version)

“In summary, dissecting our main findings per previous ZIKV-immune status we found that a ZIKV middle-convalescence: (i) results in shorter DENV viremic period, (ii) lowest pro-inflammatory status with upregulation of cellular immune response mediators, (iii) robust neutralizing antibody response higher in magnitude and durability against ZIKV strains and DENV serotypes, (iv) elevated activated and proliferating B cells, (v) early activation of cross-reactive CD4+ and CD8+ effector memory T cells, (v) and a major breadth of functional T cell response”

It is possible that the reviewer is linking “improvement” with “protection”.

On this work, we are providing evidence of improvement of the main arms of the immune response (humoral and CMI) against DENV in the presence of previous ZIKV immunity.

We are highlighting that significant differences are associated with the span of time between infections. However, this study is not directly suggesting a protective role of those differences against DENV. The conclusions are focused to an immune response improvement, and although this is not clearly protective, at least DENV pathogenesis is not exacerbated in presence of ZIKV immunity. This statement is what the title suggests.

In addition, the authors would like to have the opportunity to comment a little more on the reviewer’s asseveration that the magnitude of the immune response to flaviviruses is an unreliable marker of protective immunity.

Was the reviewer referring to the humoral or to the cellular-mediated arm of the immune response?

While the authors understand the reviewer’s point of view, we are only in partial agreement with his observation. The dynamic of the immune response, humoral and cellular, particularly to DENV is quite complex. Maybe it is one of the reasons that after more than 70 years of intense research an effective DENV vaccine is not available yet.

Authors are agree that on the humoral side the role of the magnitude in protection can be questionable due to multiple factors as the quality and the quantity of the NAbs elicited after any flaviviral infection (de Alwis et al., 2011; de Alwis et al., 2012; de Alwis et al., 2014; Wahala et al., 2009; Wahala et al., 2011). In fact, the authors and collaborators have contributed to refine the characterization of the humoral immune response to DENV using NHP as a model (Gallichotte et al., 2015).

1. de Alwis, R., et al. 2011. In-depth analysis of the antibody response of individuals exposed to primary dengue virus infection. *PLoS Negl Trop Dis* 5(6):e1188.

2. de Alwis, R., et al. 2012. Identification of human neutralizing antibodies that bind to complex epitopes on dengue virions. *Proc Natl Acad Sci U S A* 109(19):7439-44.
3. de Alwis, R., et al. 2014. Dengue viruses are enhanced by distinct populations of serotype cross-reactive antibodies in human immune sera. *PLoS Pathog* 10(10):e1004386.
4. Wahala, W. M., et al. 2009. Dengue virus neutralization by human immune sera: role of envelope protein domain III-reactive antibody. *Virology* 392(1):103-13.
5. Wahala, W. M., and A. M. Silva 2011. The human antibody response to dengue virus infection. *Viruses* 3(12):2374-95.
6. Gallichotte, E. N., et al. 2015. A New Quaternary Structure Epitope on Dengue Virus Serotype 2 Is the Target of Durable Type-Specific Neutralizing Antibodies. *MBio* 6(5):8.

On the cellular-mediated immune response, historically it has been a controversy about the positive or detrimental role of the CD8+ T cells on DENV pathogenesis, and if CD4+ T cells play any role. Currently, there is a broad consensus that an improvement in the CD8+ T cells activity may be beneficial preventing DENV infection and that the role of CD4+ T cells is more relevant during secondary DENV infections (Elong Ngonu et al., 2016; Saron et al., 2018; Tian et al., 2016; Yauch et al., 2010; Yauch et al., 2009; Zompi et al., 2012).

In addition, authors' collaborators previously found that different HLA class I and II molecules were associated with responses of different breadth and magnitude and that alleles associated with a higher magnitude of T cell responses were restricted by alleles associated with protection from severe disease (Weiskopf et al., 2013; Weiskopf et al., 2016). Despite, these positive correlations, it is apparent that a weak T cell responses do not predict disease susceptibility (Malavige et al., 2011). It is possible that interactions between CD4+, CD8+ and/or antibody responses might be responsible for this complex pattern. Alternatively, it is possible that beyond response magnitude, the specific phenotype of the responding T cells might be a key factor in determining disease. Indeed, recent studies suggest that both CD8+ and CD4+ DENV-specific T cells restricted by different HLA alleles are associated with specific phenotypes (de Alwis et al., 2016; Weiskopf et al., 2015). Currently, collaborators of the authors are working in the characterization of MHC restricted epitopes by using the data from their different works. Authors and collaborators expect to provide, in future manuscripts, insight about the role of different MHC in the breath and magnitude of the immune response.

1. Elong Ngonu, A., et al. 2016. Protective Role of Cross-Reactive CD8 T Cells Against Dengue Virus Infection. *EBioMedicine* 13:284-293.
2. Saron, W. A. A., et al. 2018. Flavivirus serocomplex cross-reactive immunity is protective by activating heterologous memory CD4 T cells. *Sci Adv* 4(7):eaar4297.
3. Tian, Y., A. Sette, and D. Weiskopf 2016. Cytotoxic CD4 T Cells: Differentiation, Function, and Application to Dengue Virus Infection. *Front Immunol* 7:531.
4. Yauch, L. E., et al. 2010. CD4+ T cells are not required for the induction of dengue virus-specific CD8+ T cell or antibody responses but contribute to protection after vaccination. *J Immunol* 185(9):5405-16.
5. Yauch, L. E., et al. 2009. A protective role for dengue virus-specific CD8+ T cells. *J Immunol* 182(8):4865-73.
6. Zompi, S., et al. 2012. Protection from secondary dengue virus infection in a mouse model reveals the role of serotype cross-reactive B and T cells. *J Immunol* 188(1):404-16.

7. de Alwis, R., et al. 2016. Immunodominant Dengue Virus-Specific CD8+ T Cell Responses Are Associated with a Memory PD-1+ Phenotype. *J Virol* 90(9):4771-4779.
8. Malavige, G. N., et al. 2011. HLA class I and class II associations in dengue viral infections in a Sri Lankan population. *PLoS One* 6(6):e20581.
9. Weiskopf, D., et al. 2013. Comprehensive analysis of dengue virus-specific responses supports an HLA-linked protective role for CD8+ T cells. *Proc Natl Acad Sci U S A* 110(22):E2046-53.
10. Weiskopf, D., et al. 2016. HLA-DRB1 Alleles Are Associated With Different Magnitudes of Dengue Virus-Specific CD4+ T-Cell Responses. *J Infect Dis* 214(7):1117-24.
11. Weiskopf, D., et al. 2015. Dengue virus infection elicits highly polarized CX3CR1+ cytotoxic CD4+ T cells associated with protective immunity. *Proc Natl Acad Sci U S A* 112(31):E4256-63.

The scenario is more complex when ZIKV and DENV immune interactions are considered. Currently, there is very few information about the protective or detrimental role of the Abs and T cells in the context of sequential infections with ZIKV and DENV (in any order). But in general, there is no a conclusive argument against the fact that the magnitude of the immune response to flaviviruses can be linked to protection.

Authors took into consideration the reviewer's suggestion on the non-detrimental effect and now the abstract reads:

"Collectively, from our work we found no evidence of a detrimental effect of ZIKV immunity in a subsequent DENV infection" (Lines 53-54 in track changes version)

2. The execution and presentation of the statistical analyses warrant substantial revision. The approach used for comparison of multiple groups (ANOVA with Tukey test) is reasonable, however, this is appropriate only if the data are normally distributed and appropriate transformations have been applied to the data. In several cases, e.g., serum viral RNA levels and neutralizing antibody titers, log transformation did not appear to be used for calculation of means and statistical comparisons.

Authors appreciate this comment that is in line with the observation of reviewer #1. As described above, in the reviewed version the viremia graph type was changed and the transformed viremia values were used (see figure above). Statistical analysis was repeated using the transformed viremia data and no significant differences were confirmed in the viremia peak among groups. Below is a table showing the Tukey multiple comparison statistical analysis of the transformed viremia data. Authors hope this may satisfy the reviewer's request.

Tukey's multiple comparisons test	Predicted (LS) mean diff.	95.00% CI of diff.	Significant?	Summary	Adjusted P Value
0					
ZIKV-immune10mo vs. ZIKV-immune2mo	0	-1.447 to 1.447	No	ns	>0.9999
ZIKV-immune10mo vs. Naive	0	-1.586 to 1.586	No	ns	>0.9999
ZIKV-immune2mo vs. Naive	0	-1.447 to 1.447	No	ns	>0.9999
1					
ZIKV-immune10mo vs. ZIKV-immune2mo	-0.9587	-2.406 to 0.4888	No	ns	0.2622
ZIKV-immune10mo vs. Naive	-0.2789	-1.865 to 1.307	No	ns	0.9087
ZIKV-immune2mo vs. Naive	0.6798	-0.7677 to 2.127	No	ns	0.5076
2					
ZIKV-immune10mo vs. ZIKV-immune2mo	-0.1242	-1.572 to 1.323	No	ns	0.9775
ZIKV-immune10mo vs. Naive	-0.4303	-2.016 to 1.155	No	ns	0.7965
ZIKV-immune2mo vs. Naive	-0.3061	-1.754 to 1.141	No	ns	0.8708
3					
ZIKV-immune10mo vs. ZIKV-immune2mo	-0.6923	-2.140 to 0.7552	No	ns	0.4951
ZIKV-immune10mo vs. Naive	-0.7152	-2.301 to 0.8704	No	ns	0.5349
ZIKV-immune2mo vs. Naive	-0.02287	-1.470 to 1.425	No	ns	0.9992
4					
ZIKV-immune10mo vs. ZIKV-immune2mo	-0.02294	-1.470 to 1.425	No	ns	0.9992
ZIKV-immune10mo vs. Naive	0.228	-1.358 to 1.814	No	ns	0.938
ZIKV-immune2mo vs. Naive	0.251	-1.197 to 1.698	No	ns	0.9112
5					
ZIKV-immune10mo vs. ZIKV-immune2mo	0.1727	-1.275 to 1.620	No	ns	0.9569
ZIKV-immune10mo vs. Naive	0.6808	-0.9049 to 2.266	No	ns	0.567
ZIKV-immune2mo vs. Naive	0.5081	-0.9394 to 1.956	No	ns	0.6838
6					
ZIKV-immune10mo vs. ZIKV-immune2mo	0.08971	-1.358 to 1.537	No	ns	0.9882
ZIKV-immune10mo vs. Naive	0.7514	-0.8342 to 2.337	No	ns	0.5015
ZIKV-immune2mo vs. Naive	0.6617	-0.7858 to 2.109	No	ns	0.5259
7					
ZIKV-immune10mo vs. ZIKV-immune2mo	-0.2377	-1.685 to 1.210	No	ns	0.9199
ZIKV-immune10mo vs. Naive	-0.6587	-2.244 to 0.9269	No	ns	0.5878
ZIKV-immune2mo vs. Naive	-0.421	-1.869 to 1.026	No	ns	0.77
8					
ZIKV-immune10mo vs. ZIKV-immune2mo	-0.08332	-1.531 to 1.364	No	ns	0.9898
ZIKV-immune10mo vs. Naive	-1.389	-2.974 to 0.1969	No	ns	0.0988
ZIKV-immune2mo vs. Naive	-1.305	-2.753 to 0.1421	No	ns	0.0862
9					
ZIKV-immune10mo vs. ZIKV-immune2mo	-0.1372	-1.585 to 1.310	No	ns	0.9726
ZIKV-immune10mo vs. Naive	-1.036	-2.622 to 0.5494	No	ns	0.2716
ZIKV-immune2mo vs. Naive	-0.899	-2.346 to 0.5485	No	ns	0.3076
10					
ZIKV-immune10mo vs. ZIKV-immune2mo	0	-1.447 to 1.447	No	ns	>0.9999
ZIKV-immune10mo vs. Naive	0	-1.586 to 1.586	No	ns	>0.9999
ZIKV-immune2mo vs. Naive	0	-1.447 to 1.447	No	ns	>0.9999
15					
ZIKV-immune10mo vs. ZIKV-immune2mo	0	-1.447 to 1.447	No	ns	>0.9999
ZIKV-immune10mo vs. Naive	0	-1.586 to 1.586	No	ns	>0.9999
ZIKV-immune2mo vs. Naive	0	-1.447 to 1.447	No	ns	>0.9999

Related to the neutralization data, the original statistical analysis was ran using the transformed neutralization results and plotted in Supplementary Figures 4 and 6.

- P values are provided throughout the text for the post-hoc pairwise comparisons to 4 significant digits, and the authors suggest that smaller p values indicate more biologically significant findings. This needs to be reconsidered, particularly in light of the small sample sizes (4-6 per group).

Authors understand the reviewer's rationale on this particular issue. However, in an ample review on the use of NHP as a model for DENV, it was established that using groups of 4 to 6 animals have yielded interpretable data (Sariol and White, 2014). A critical factor determining the number of animals needed is related to the genetic background of the animals (Sariol and White, 2014). Our group has access to a unique population of rhesus macaques having the highest homogenous genetic background (more than 90% Indian-origin ancestor) of all rhesus populations in the USA (Kanthaswamy et al., 2016; Kanthaswamy et al., 2018; Kanthaswamy et al., 2010). This population, with very limited variability from animal to animal, allow to use the suggested smaller number of animals per group with the potential to obtain interpretable data.

In addition, the experimental design (number of animals per group) and statistical analyses supporting the results of this work are in agreement with most of the published studies on DENV/ZIKV field, which include precisely 4 to 6 animals per group and provided comparable statistical analyses (George et al., 2017; Lum et al., 2018; Osuna et al., 2016; Pantoja et al., 2017; Valiant et al., 2018).

1. Sariol, C. A., and L. J. White 2014 Utility, limitations, and future of non-human primates for dengue research and vaccine development. *Front Immunol* 5:452.
2. Kanthaswamy, S., et al. 2016 The Population Genetic Composition of Conventional and SPF Colonies of Rhesus Macaques (*Macaca mulatta*) at the Caribbean Primate Research Center. *J Am Assoc Lab Anim Sci* 55(2):147-51.
3. Kanthaswamy, S., et al. 2018 Determination of major histocompatibility class I and class II genetic composition of the Caribbean Primate Center specific pathogen-free rhesus macaque (*Macaca mulatta*) colony based on massively parallel sequencing. *J Med Primatol* 47(6):379-387.
4. Kanthaswamy, S., et al. 2010 Detecting signatures of inter-regional and inter-specific hybridization among the Chinese rhesus macaque specific pathogen-free (SPF) population using single nucleotide polymorphic (SNP) markers. *J Med Primatol* 39(4):252-65.
5. George, J., et al. 2017 Prior Exposure to Zika Virus Significantly Enhances Peak Dengue-2 Viremia in Rhesus Macaques. *Sci Rep* 7(1):10498.
6. Lum, F. M., et al. 2018 Multimodal assessments of Zika virus immune pathophysiological responses in marmosets. *Sci Rep* 8(1):17125.
7. Osuna, C. E., et al. 2016 Zika viral dynamics and shedding in rhesus and cynomolgus macaques. *Nat Med* DOI 10.1038/nm.4206.
8. Pantoja, P., et al. 2017 Zika virus pathogenesis in rhesus macaques is unaffected by pre-existing immunity to dengue virus. *Nat Commun* 8:15674.
9. Valiant, W. G., et al. 2018 Zika convalescent macaques display delayed induction of anamnestic cross-neutralizing antibody responses after dengue infection. *Emerg Microbes Infect* 7(1):130.

- Finally, the authors should review all figures to ensure that individual data points or an estimate of the variance has been provided wherever appropriate.

Authors are agree, an overall review of all figures was performed.

3. The authors have not addressed the limitations of their study; this should be included in the Discussion. Overall, the Discussion is excessively long, repeats background information addressed in the Introduction, and is overly speculative given the limited differences observed in the study.

Limitations of the study had been added to the reviewed version (Lines 1047-1057 in track changes version).

One limitation of our study is the utilization of low numbers of animals per group. Additional studies with a larger number of animals are warranted. However, fundamental and seminal contributions on ZIKV and ZIKV/DENV interactions have been obtained by using similar limited number of animals per group (Dudley et al., 2017; George et al., 2017; Magnani et al., 2017; Masel et al., 2019; McCracken et al., 2017; Osuna et al., 2016; Pantoja et al., 2017; Serrano-Collazo et al., 2019; Silveira et al., 2017). Another limitation is that our study monitored the immune response up to 90 days after DENV infection. Additional longitudinal studies are needed to test the immune response over longer periods of time including subsequent DENV heterotypic challenges to evaluate the efficacy of the memory recall in cross-protection between serotypes. Finally, we cannot comment about the likelihood to increase or decrease susceptibility to develop DHF/DSS in the context of ZIKV immunity since DENV clinical manifestations in NHP models are limited and are characterized to be subclinical infections (Sariol and White, 2014).

1. Dudley, D. M., et al. 2017. Infection via mosquito bite alters Zika virus tissue tropism and replication kinetics in rhesus macaques. *Nat Commun* 8(1):2096.
2. George, J., et al. 2017. Prior Exposure to Zika Virus Significantly Enhances Peak Dengue-2 Viremia in Rhesus Macaques. *Sci Rep* 7(1):10498.
3. Magnani, D. M., et al. 2017. Neutralizing human monoclonal antibodies prevent Zika virus infection in macaques. *Sci Transl Med* 9(410).
4. Masel, J., et al. 2019. Does prior dengue virus exposure worsen clinical outcomes of Zika virus infection? A systematic review, pooled analysis and lessons learned. *PLoS Negl Trop Dis* 13(1):e0007060.
5. McCracken, M. K., et al. 2017. Impact of prior flavivirus immunity on Zika virus infection in rhesus macaques. *PLoS Pathog* 13(8):e1006487.
6. Osuna, C. E., et al. 2016. Zika viral dynamics and shedding in rhesus and cynomolgus macaques. *Nat Med* DOI 10.1038/nm.4206.
7. Pantoja, P., et al. 2017. Zika virus pathogenesis in rhesus macaques is unaffected by pre-existing immunity to dengue virus. *Nat Commun* 8:15674.
8. Serrano-Collazo, Crisanta, et al. 2019. Significant control of Zika infection in macaques depends on the elapsing time after dengue exposure. *bioRxiv*:625293.
9. Silveira, E. L. V., et al. 2017. Immune Cell Dynamics in Rhesus Macaques Infected with a Brazilian Strain of Zika Virus. *J Immunol* 199(3):1003-1011.

10. Sariol, C. A., and L. J. White 2014. Utility, limitations, and future of non-human primates for dengue research and vaccine development. *Front Immunol* 5:452.

The discussion has been edited to be more concise. However, we are in partial agreement with the comment about “overly speculative”.

In contrast to other NHPs studies on ZIKV and DENV interactions that are based mostly on trends due to the high variability within groups and lower number of animals, we believe that our study provides a reasonable piece of evidence with statistical significance or important trends supporting our statements. Moreover, cross-immunological data on this scenario (ZIKV-DENV) is scarce in large animal models which are comparable to humans such as NHP. We have raised the possibility that our findings may explain what will occur immunologically in human population in countries with ZIKV-DENV herd immunity.

The nature of previously published studies very accepted in the field may be also considered speculative. Studies that are based on cases count and serological data (not necessarily including information about pathogenesis) from a specific region or country, often extrapolate this through simulation and modeling to anticipate what may occur epidemiologically in the future. An example of this type of study, also under consideration in NatComms:

<https://nature-research-under-consideration.nature.com/users/37265-nature-communications/posts/45833-dengue-after-zika-characterizing-impacts-of-zika-emergence-on-endemic-dengue-transmission>

Another example of extrapolating serological results to advance epidemiological or global concepts can be found in a recent article by Rodriguez-Barraquer et al.

1. *Rodriguez-Barraquer, I., et al. 2019. Impact of preexisting dengue immunity on Zika virus emergence in a dengue endemic region. Science 363(6427):607-610.*

4. The comparison of 2 mo and 10 mo intervals between ZIKV and DENV challenge is a major focus of the authors' conclusions. While the description of the findings in these two groups is accurate, the authors do not adequately address differences between these groups at baseline (time of DENV challenge) and alternative explanations for their findings. Some additional methodological details are essential. Did the two different ZIKV challenge strains perform similarly- e.g., duration and magnitude of viremia, magnitude of antibody titers at equivalent time points, etc.? This is particularly relevant since figure 5 shows higher neutralizing antibody titers to ZIKV at 10 mo post-infection in the ZIKV_{PF}-10mo group than at 2 mo post-infection in the ZIKV_{PR}-2mo group and figure S2 shows higher anti-ZIKV NS1 IgG; differences in the initial immune response to ZIKV (despite equivalent challenge doses as PFU) would potentially explain the differences observed after DENV challenge.

Authors are very glad to see that this point was brought up. There are very limited data comparing these two strains in terms of replication and magnitude of the neutralizing antibodies they induce. Precisely, this is the focus of another manuscript in preparation by the authors. Authors are happy to share some of these relevant data. Authors had found that H/PF/2013 ZIKV strain replicates more efficiently in rhesus. To our knowledge, there are only two studies (from the authors) supporting this observation.

1. *Pantoja, P., et al. 2017. Zika virus pathogenesis in rhesus macaques is unaffected by pre-existing immunity to dengue virus. Nat Commun 8:15674.*

2. Serrano-Collazo, Crisanta, et al. 2019. Significant control of Zika infection in macaques depends on the elapsing time after dengue exposure. *bioRxiv:625293*.

The figure below shows the replication differences between the H/PF/2013 and the PRVABC59 ZIKV strains in rhesus macaques and the impact of previous DENV immunity.

(A) ZIKV viremia and (B) average viremia days in DENV-immune rhesus macaques. Animals with 2.8 years of DENV-immunity were challenged with the PF/H/2013 ZIKV strains. Animals with 12mo or 3mo of previous DENV-immunity were challenged with the PRVABC59 ZIKV strain. As shown in panel A, the PRVABC59 strain replicates in rhesus macaques ~1-2 logs lower than the H/PF/2013. However, regardless of the infecting strain, in all cases previous DENV-immunity results in a decrease in the average ZIKV viremia days. Significant differences were only observed for the group with 12mo of DENV-immunity.

These results on the differences in replication properties between those two ZIKV strains confirm the data provided by the BEI in the package insert of the PRVABC59 ZIKV strain.

[redacted]

However, in spite of higher replication of the H/PF/2013 strain, the NAb response reach similar titers at 30 and 60 after ZIKV infection.

Neut60 Ab Titers vs ZIKV H/PF/2013 & PRVABC59
After ZIKV infection

Neutralizing titers induced by two different ZIKV strains in rhesus macaques. The magnitude of the NAbs induced by both ZIKV strains at 30 and 60 dpi is comparable regardless the higher replication efficiency of the H/PF/2013 strain.

This figure is now provided as Supplementary Figure 6 and described under results section (Lines 456-461 in track changes version)

These results support that the differences reported in this work, including the differences at baseline between ZIKV-immune groups, are most likely associated to the time elapsed between ZIKV and DENV infections and are unaffected by the fitness of the pre-infecting strain. In addition, these results reinforce the homogeneous efficiency of NAbs against multiple ZIKV strains (Dowd et al., 2016; Collins et al., 2017; Swanstrom et al., 2016).

1. Dowd, K. A. et al. Broadly Neutralizing Activity of Zika Virus-Immune Sera Identifies a Single Viral Serotype. *Cell reports* **16**, 1485-1491, doi:10.1016/j.celrep.2016.07.049 (2016).
2. Collins, M. H. et al. Lack of Durable Cross-Neutralizing Antibodies Against Zika Virus from Dengue Virus Infection. *Emerging infectious diseases* **23**, 773-781, doi:10.3201/eid2305.161630 (2017).
3. Swanstrom, J. A. et al. Dengue Virus Envelope Dimer Epitope Monoclonal Antibodies Isolated from Dengue Patients Are Protective against Zika Virus. *mBio* **7**, DOI 10.1128/mBio.01123-01116, doi:10.1128/mBio.01123-16 (2016).

- Were all three cohorts challenged with DENV at the same time, so that assays on fresh PBMC were done concurrently? (This latter point is critical to the interpretation of some differences in flow cytometry data at baseline, such as CD69 staining shown in figure 6, that are otherwise difficult to understand.)

Due to complex logistic issues working with NHP the 2mo cohort was challenged 3 months apart from the other two cohorts. Similar approach (challenging group of animals at different time points) has been used for previous experimental designs without evidence of bias in the outcome. Authors consider that the data provided in the manuscript support that the differences among groups, such as CD69 staining of the T-EM and T-CM cells, were not due to the timing of the challenge. This is strongly supported by the fact that the % of total lymphocytes, % of total CD4+ and CD8+ T cells were very similar among all three groups. These similarities support the lack of bias in the differences observed in the activation (CD69+) of T cell subpopulations since it is not due by heterogeneous frequency of cell subsets. In addition, the frequency of activated CD4+ T cells using the same marker (CD69) tend to be higher in the 2mo group, so this group also had a productive activation of the T cell response. On the other hand, the frequency of activated CD8+ T cells remains very similar for all three groups (Now Supplementary figure 13). Additional arguments supporting the validity of comparing results obtained from NHP challenged at different time points are as follow:

1-Other groups working with NHP use what it is called historical controls for comparison when all conditions are the same. For example:

Aliota MT, Dudley DM, Newman CM, Mohr EL, Gellerup DD, et al. (2016) Heterologous Protection against Asian Zika Virus Challenge in Rhesus Macaques. *PLOS Neglected Tropical Diseases* 10(12): e0005168. <https://doi.org/10.1371/journal.pntd.0005168>

2-Same method has been in use in the laboratory for many years to challenge the animals (including the same Veterinarian).

3-Use of same DENV viral stock.

4-Phenotyping analyses were conducted using fresh PBMCs within the first 24 hours of collection. For this we use the same reagents, same device, the same person (Pérez-Guzmán E.X., first author), and same software.

5-The way these experiments were conducted provided independent replicates which make the stats more robust.

6-Finally but not least, every time a group working with any animal model is requested to repeat an experiment in order to provide additional data, the infection is conducted at a different time point.

- How were groups “matched” for age and sex of animals, especially when the sample sizes were not identical? If groups were matched at the time of DENV challenge, could the 8 mo difference in ages at the time of ZIKV challenge have contributed to differences between the groups? (The ages and sexes of each animal at the time of DENV challenge could be included in the supplementary material.)

Historically authors have used the same description of our cohorts. The term “matched” referred only to sex and age. This is why we specified those terms. However following reviewer observation, we modified the text in the manuscript and now it reads:

“The ages of all animals are within the age range for young adults rhesus macaques <https://www.nc3rs.org.uk/macques/macques/life-history-and-diet/> (ZIKVPF-10mo: 6.8, 6.8, 5.8, and 5.9; ZIKVPR-2mo: 6.4, 6.5, 5.2, 4.3, 5.6, and 5.5; Naïve: 4.8, 6.6, 6.8, and 5.7).” (Now lines 1222-1228 in track changes version)

For the reviewer’s benefice, authors want to comment that those age ranges are all on the young adult side, when the highest reproductive output for all females on the CPRC population is observed for example (from 4-8yrs old). Below some references that can be reviewed on this particular topic of age:

<https://www.nc3rs.org.uk/macques/macques/life-history-and-diet/>. (This reference is particularly interesting and easy to review),
<http://www.sunypress.edu/p-83-the-cayo-santiago-macaques.aspx>
and
<https://link.springer.com/article/10.1007/BF00164042>

This new graph was added as panel k to Supplementary figure 1: **Clinical status and vital signs kinetics in ZIKV-immune and naïve macaques.**

5. The authors found no significant differences in viremia between groups. This supports their conclusion that they did not observe an enhancement of viremia, and their discussion of potential explanations for the difference from published data is adequate.

Authors appreciate the reviewer found adequate the discussion on this particular issue.

- It is also fair to point out trends toward lower viremia at days 1 and 8, given the low statistical power of the study, but the authors overinterpret these data and speculate excessively in the Discussion about potential mechanisms for a non-significant difference.

Authors are in agreement with the reviewer's observation. The following text making a link between viremia and CD8 T cells activation was removed:

"Since CD8+ T cells are known for their role in early clearance of DENV in heterologous secondary infections⁶⁸, it is possible that the T cell cross-reactivity with DENV E detected in the ZIKV mid-convalescent group at baseline together with the basal levels of cross-neutralizing Abs against DENV-2 before DENV infection, provide a synergistic and partial cross-protective effect:

- Data are also unnecessarily duplicated between figure 2 and table S2. Figure S4 re-analyzes the viremia and neutralizing antibody data; although the authors present this analysis (lines 313-334) as identifying an inverse correlation between these parameters, the analysis is inadequate to support this conclusion- mean values are shown for each group without a correlation analysis at the level of individual animals, and no statistical tests are applied to the data. Figure S4 should be removed or a more detailed analysis should be conducted to address the authors' hypothesis.

We agree with the reviewer comment on the value and duplication in those figures.

In the reviewed version a new viremia figure 2 is provided. Also an AUC figure was added as Supplementary Figure 2 following reviewer's 1 analysis and recommendation.

In addition, Supplementary Figure 4 and Supplementary table 2 were removed.

6. The authors found higher IFN- α and IL-6 levels in the ZIKV-naïve group and higher perforin and CXCL10 levels in the ZIKVVPF-10mo group. The overall description of these data, however, is somewhat inaccurate. IFN- α would not usually be included among the pro-inflammatory cytokines, for example. More significantly, the authors did not analyze levels of those cytokines reported by George et al to be increased in ZIKV-immune animals, IL-8 and IP-10, and as a result cannot compare results. This should be noted in the Discussion.

We appreciate this comment and the opportunity to address this. Although IFN- α may not be usually considered as a pro-inflammatory cytokine (while some works in the literature do), it acts as a marker for elevated DENV replication, since a high number of viral particles activate the cellular IFN- α/β pathway to reach the antiviral state. IFN- α production indirectly contributes to explain enhanced pathogenesis which is usually accompanied by the pro-inflammatory cytokine storm. Elevated levels of IFN- α have been correlated with severity in DHF patients. Nevertheless, as requested for the reviewer, we will not describe IFN- α as a pro-inflammatory cytokine, but a marker of DENV disease severity.

This information was added in the manuscript (Lines 1031-1034 in track changes version).

1. Kurane, I. et al. High levels of interferon alpha in the sera of children with dengue virus infection. *Am J Trop Med Hyg* 48, 222-229 (1993).
2. Singla, M. et al. Immune Response to Dengue Virus Infection in Pediatric Patients in New Delhi, India--Association of Viremia, Inflammatory Mediators and Monocytes with Disease Severity. *PLoS neglected tropical diseases* 10, e0004497, 1199 doi:10.1371/journal.pntd.0004497 (2016).

While authors does not use exactly the same cytokine profile as George et al, the proinflammatory profile can actually be discussed and supported by the cytokines we measured (IL-6, MIG, MCP1, MIP-1B, IL-1RA). In addition, IP-10, also called CXCL10, that was analyzed by George et al. (as per the reviewer comment), is included in this work (see original Figure 3g). CXCL10 is an immune mediator for T cells proliferation, recruitment of CD4+ and CD8+ activated T cells and IFN- γ -producing CD8+ T cells, required to control DENV infection in vivo. The levels of this chemokine correlate with a more dynamic T cell response in the ZIKVVPF-10mo group in this work. In George et al., the authors overlooked the role of IP-10/CXCL10 in the T cell activation and T cell chemoattraction, which was expected since the contribution of the T cell response was not addressed in that study.

1. Dufour, J. H. et al. IFN-gamma-inducible protein 10 (IP-10; CXCL10)-deficient mice reveal a role for IP-10 in effector T cell generation and trafficking. *J Immunol* 168, 3195-3204 (2002).
2. Hsieh, M. F. et al. Both CXCR3 and CXCL10/IFN-inducible protein 10 are required for resistance to primary infection by dengue virus. *J Immunol* 177, 1855-1863 (2006).
3. Dejnirattisai, W. et al. A complex interplay among virus, dendritic cells, T cells, and cytokines in dengue virus infections. *J Immunol* 181, 5865-5874 (2008).

Minor comments:

7. Line 70- The statement that DENV possesses one of the highest mortality rates among arboviruses is inaccurate.

This comment was corrected. Now reads:

“DENV is a global public health threat, having two-thirds of world’s population at risk of infection, causing ~390 million infections annually”. (Lines 75-77 in track changes version)

8. Figure S2- The time points after ZIKV infection in panel e should be explained in the legend.

This description was added to the figure legend, now Supplementary Figure 3.

“Panel e includes additional timepoints before DENV infection for ZIKV-immune groups: 30, 60, 90 and 180 days after ZIKV (H/PF/2013) infection for the ZIKVVPF-10mo group, and 30 days after ZIKV (PRVABC59) infection for the ZIKVPR-2mo group.”

9. Line 294- Although neutralizing antibody titers remained higher in the ZIKVVPF-10mo group, the rate of decay appears to be identical to the ZIKVPR-2mo group.

Expected for both ZIKV-immune groups, ZIKV cross-neutralization will decrease overtime by the stimulation of DENV infection. However, the ZIKVVPF-10mo group retained higher NAb titers against ZIKV until the end of the study as consequence of the higher titers peak between 15-30 dpi.

However the sentence was edited and now reads:

“In general, the neutralizing response of the ZIKVVPF-10mo group maintained higher NAb titers up to 90 dpi compared to ZIKVPR-2mo and naïve groups”. (Lines 346-348 in track changes version)

10. Line 338- The meaning of heterologous ZIKV strains in ZIKV-immune groups is unclear. Also, the authors cannot draw conclusions about long-lasting antibody responses in the ZIKVPR-2mo group.

Agree. The terms “heterologous” and “long-lasting” were removed from the sentence. Now reads: “Previous exposure to ZIKV strains in ZIKV-immune groups developed high levels of cross-reactive, non-neutralizing, and neutralizing Abs before DENV infection (baseline)”. (Lines 379-381 in track changes version)

11. Figure 5b- The authors do not provide sufficient context for why they analyzed EC50 levels. In addition, it is unclear if/how this differs from PRNT50 calculations, and the symbols in the chart are not defined in the legend.

Agree. The EC50 data is repetitive. Figure 5b was eliminated accordingly.

12. Figure S5- Neutralization curves for days 15-60 likely do not allow reliable modeling of PRNT60.

Authors are agree with this observation. However, while we did not expanded the dilutions to show the endpoint dilutions for every single animal, the figure fulfill its role as supplementary figure to show the change in behavior of the sigmoidal curves by group overtime. The results from each

animal independently on this figure confirm the main results summarized in Figure 5a, which supports a higher ZIKV neutralizing magnitude in the ZIKVVPF-10mo group compared to the other two groups.

In addition, on previous Figure S5 (Now Figure S6) and Figure S3 (Now Figure S4) the labels at the top for each ZIKV-immune group were edited from “ZIKV-10mo and ZIKV-2mo” to “ZIKVVPF-10mo and ZIKVPR-2mo”.

13. Representative flow cytometry and gating strategies should be provided for the analyses of DC and NK cells.

Representative gating strategies for DCs and NK cells are now provided as Supplementary Figures 8 and 9. Details about the gating and selected populations are described in the methods section: Immunophenotyping.)

14. Figure S8 and line 391- NK8 and NK56 should be CD8 and CD56.

Correct, this has been corrected in the results and methods sections and in now Supplementary Figure 11, labels now read: NKCD8 and NKCD56.

15. Lines 401 and 410- The statistical analyses performed did not test for differences between groups.

Agree, Dunnett test does not compare differences between groups, it compares differences within a group (each individual timepoint vs baseline of the same group). This has been re-worded from “between groups” to “within groups” in the results section of frequency of immune cell subsets. The correct definition of Dunnett test comparison is now available in each immunophenotyping main or supplementary figure legend.

Authors opted to perform Dunnett instead of Tukey test for immunophenotyping since we observed heterogeneous levels of immune cell subsets between groups at baseline that not necessarily are related to previous ZIKV memory immunity. Immunophenotyping measured the frequency of non-specific cell subsets, which may be difficult to attribute a direct role in DENV infection of these cells. The objective was to detect significant differences of cell subsets elicited by DENV infection regardless of their basal levels. A cutted line divide percent of frequency measured before and after DENV infection in all panels related to immunophenotyping. Although not specific, significant increases are more related to DENV infection and may have an effect in the course of infection. In contrast, for PRNT we use Tukey test since NAb titers come from ZIKV and/or DENV type-specific or cross-reactive Ab-producing B cells and baseline levels of this may have a direct impact in the outcome of DENV infection.

16. Line 411-413- The statement regarding B cell phenotyping data is accurate for the ZIKVVPF-10mo group but not for the ZIKVPR-2mo group.

This sentence was corrected and now reads (Lines 553-555 in track changes version):

“Together, these phenotyping results of B cells are consistent with the early and boosted production of binding and neutralizing Abs in the ZIKV-10mo group compared to naïve animals”

17. Figure S6- The gating of EM, CM, and naïve subpopulations is incorrectly labeled.

The labeling for EM (CD95+CD28-), CM (CD95+CD28+) and Naïve (CD95-CD28+) T cell subpopulations was corrected in now Supplementary Figure 7.

18. Figure S10- The percentage of CD69+ cells in total T cells does not appear to be consistent with the results in CD4 and CD8 subsets (e.g., ZIKVPR-2mo group). The order of symbols differs in panel d- is this correct?

Authors understand the reviewer’s point of view. Normally, total T cells exhibit low levels of activation. However, it is expected that when total T cells are divided in both CD4+ and CD8+ compartments the percent of activated cells will be higher since the absolute number of cells in the selected population is lower. This means that in a small amount of cells few activated cells will represent a higher percent from the small total amount of cells. As reviewer noted, the ZIKVPR-2mo group is a good example since it have the lowest activated total T cells, but the percent within T cell compartments is inflated, which means that the majority of those few activated total T cells are activated within the compartments.

Yes. The order of the symbols is now correct in panel d.

19. The authors tested for T cell responses to ZIKV NS1 peptides and not to other non-structural proteins. This should be noted clearly in the text.

Thank you for this observation. Actually, the experiments were conducted using a pool of peptides from all ZIKV NS proteins. The supplementary table was incomplete showing NS1 peptides only. The Supplementary table 4 is now completed including peptide libraries for each NS protein.

20. All plots of antigen-specific T cell responses show the frequencies of cells expressing individual effector functions; the authors do not present data on polyfunctional responses, i.e., expression of multiple effector functions by individual T cells. The text should be clarified or the data on such responses should be included, at least in summary form.

Authors are agree with the reviewer, the data shows individual effector functions. We clarified this by changing “Polyfunctional response” to “Functional response” in the text and figure legend. However, our initial definition of “Polyfunctional” was that although the data do not show multiple effector functions at the same time, collectively these animals are producing an overall T cell response that is capable to produce/express several effector molecules. However, we are completely agree to perform this change suggested by the reviewer, since may be confusing for the readers as well.

21. Lines 556-557- There is inadequate data to support this statement, and certainly not to draw a strong conclusion.

Authors believe that they have convincing data suggesting that the dynamic of the immune response is different in the two scenarios of sequential infections DENV-ZIKV or ZIKV-DENV. However, authors deleted that statement as part of the shortening in the discussion section.

22. Lines 575-576- The data presented in figure 5c do not support this statement.

This statement: “Altogether, these results demonstrate that DENV infection results in a significant increase in the magnitude and durability of the cross-neutralizing Ab response against ZIKV in animals with a mid-convalescent period from ZIKV infection” is one of the main findings of this work.

Authors may agree with the reviewer in that the information provided in figure 5c may not be enough to support this statement. However, this is now reinforced with the addition of the new data (provided in response major comment 4) showing in parallel the similarity in the NAbs titers against ZIKV strains at 30 and 60 days after ZIKV infection.

As stated above, these results support that the differences reported in this work, including the differences at baseline between ZIKV-immune groups, are more likely associated to the time elapsed between ZIKV and DENV infections—explained by the maturation of ZIKV immune memory and the subsequent recall by DENV infection—and are unaffected by the fitness of the pre-infecting strain.

23. Line 839- Animals were (likely) not bled continuously.

Animals were continuously bled from day 1 to 10 as described, and after that on days 15, 30, 60 and 90 post-infection. The clarification “after that” was inserted in the text.

24. Line 947- The authors should clarify the number of comparisons used for adjustment.

Authors clarify the number of comparisons used for adjustments in all figure legends.

Reviewers' Comments:

Reviewer #1:

Remarks to the Author:

The authors responded properly to the suggestions and comments of the reviewers.

However, the text still has some grammatical/syntax errors (including some new ones due to the revision), which should be corrected for the final publication.

One minor suggestion in reference to a prior question by another reviewer:

- On line 1235, change "continuously" with "once daily", as authors mean that animals were bled once daily, and continuously may give the false impression that an IV catheter was used for ongoing blood collection.

Reviewer #3:

Remarks to the Author:

I feel that the following points have NOT been adequately addressed in the revised manuscript:

1. Improvement of immune responses- The authors have retained this terminology in the title, abstract, and critical locations in the manuscript. Their lengthy rebuttal in the main agrees with the criticism that there are no reliable criteria for a "better" immune response to DENV, and that magnitude alone is insufficient. However, the manuscript clearly wishes to equate the two, and this leads to a misleading title and abstract. Similar higher antibody and T cell responses are observed, for example, when DENV-immune animals are challenged with a heterologous DENV, yet those would not usually be considered "improved". Furthermore, the papers cited to support the authors' interpretation mainly refer to immune responses measured prior to DENV challenge, whereas the authors have measured higher immune responses AFTER clearance of viremia; at a minimum, further clarification would be needed to specify and justify what positive outcome the authors anticipate as a result of the effect observed.

2. Comparison of 2 mo and 10 mo intervals- I appreciate the inclusion of data on neutralizing antibody titers to ZIKV at 30 and 60 d after ZIKV infection and other revisions to address this point. Although the neutralizing antibody data support the authors' interpretation that time post-ZIKV infection best explained the differences in outcomes, it is problematic that the authors have chosen not to report other known potential sources of systematic bias. The significant difference in viremia from these two strains should at least receive comment in the Discussion. Also, the Methods section should note that the ZIKVPR-2mo cohort was challenged separately from the other two cohorts. This is clearly a potential source of bias, particularly for assays done on fresh blood samples. The authors are entitled to explain why they do not believe this explains differences between groups but should not keep this pertinent information from the reader.

3. Statistical analyses- The authors appear to have missed the point that reporting p value estimations to 4 significant digits suggests a precision that is not typically considered appropriate for such small samples. Their comments (in the rebuttal and the discussion) citing other papers using similar group sizes are not pertinent to this point, but I would yield to the usual practice of the journal. Additional points regarding specific figures:

a. Figure 2a- The meaning of the box and whiskers should be specified.

b. Figure 2b- There is no information on the distribution of values within each group.

c. Figure S2- The Methods do not indicate what value was imputed for samples below LOD in

calculation of means.

4. Limitations of the study- I appreciate the authors' effort to address this point. A point that should be considered is that the study only included two specific time points after ZIKV infection, 2 and 10 months. The authors do not address the appropriateness of selecting these two time points and the potential for different findings at other biologically relevant intervals.

5. Viremia data- I appreciate the authors' clarifications. Regarding the new analysis based on AUC, it would be more informative to calculate this value separately for each animal and compare groups using appropriate statistical tests. (In this case, the authors should be sure to note that this was a post-hoc analysis.)

6. Cytokine levels- I believe it would be helpful to the readers for the authors to have noted which specific cytokines were found elevated in the study by George et al.

7. Figure S13 (formerly S10)- The authors response does not allay my specific point of concern. Comparing the data for the ZIKVPR-2mo group between panels d, e, and f, most values for the percentages of CD69+ cells in total T cells appear to be <5%, whereas corresponding values for CD4 and CD8 subsets often range over 20%. The authors were directed to these findings specifically because normally these two subsets would encompass nearly all T cells. The authors' explanation wouldn't be adequate unless there was a substantial percentage of T cells that did not fall into either subset. Was that the case, and, if so, what is the authors' explanation?

Reviewers' comments on the re-submission of June 24th, 2019.

Manuscript Number: NCOMMS-19-12662A

Authors' answers in *italics*

(For Lines refer to Clean version of the revised manuscript)

Reviewer #1 (Remarks to the Author):

The authors responded properly to the suggestions and comments of the reviewers.

However, the text still has some grammatical/syntax errors (including some new ones due to the revision), which should be corrected for the final publication.

An extensive grammatical/syntax review was completed.

One minor suggestion in reference to a prior question by another reviewer:

- On line 1235, change "continuously" with "once daily", as authors mean that animals were bled once daily, and continuously may give the false impression that an IV catheter was used for ongoing blood collection.

"Continuously" was changed to "once daily".(Line 786)

Reviewer #3 (Remarks to the Author):

I feel that the following points have NOT been adequately addressed in the revised manuscript:

1. Improvement of immune responses- The authors have retained this terminology in the title, abstract, and critical locations in the manuscript. Their lengthy rebuttal in the main agrees with the criticism that there are no reliable criteria for a "better" immune response to DENV, and that magnitude alone is insufficient. However, the manuscript clearly wishes to equate the two, and this leads to a misleading title and abstract. Similar higher antibody and T cell responses are observed, for example, when DENV-immune animals are challenged with a heterologous DENV, yet those would not usually be considered "improved". Furthermore, the papers cited to support the authors' interpretation mainly refer to immune responses measured prior to DENV challenge, whereas the authors have measured higher immune responses AFTER clearance of viremia; at a minimum, further clarification would be needed to specify and justify what positive outcome the authors anticipate as a result of the effect observed.

On this new version we re-write parts of the abstract and discussion avoiding the term of "improvement" and added some new comments comparing the ZIKV-DENV scenario with consecutive DENV heterologous infections in terms of the magnitude of the responses. We believe this new approach highlighting an essential information that was not discussed previously is in line with the reviewer's comments and point of view.

Abstract: (Lines 42-45)

Discussion: (Lines 492-495; 539-557)

In addition, to avoid confusion we modified the title:

"Time elapsed between Zika and dengue virus type 2 infections alters the magnitude of antibody and T cell responses but not viremia in rhesus macaques"

2. Comparison of 2 mo and 10 mo intervals- I appreciate the inclusion of data on neutralizing antibody titers to ZIKV at 30 and 60 d after ZIKV infection and other revisions to address this point. Although the neutralizing antibody data support the authors' interpretation that time post-ZIKV infection best explained the differences in outcomes, it is problematic that the authors have chosen not to report other known potential sources of systematic bias.

The significant difference in viremia from these two strains should at least receive comment in the Discussion.

Also, the Methods section should note that the ZIKVPR-2mo cohort was challenged separately from the other two cohorts. This is clearly a potential source of bias, particularly for assays done on fresh blood samples.

The authors are entitled to explain why they do not believe this explains differences between groups but should not keep this pertinent information from the reader.

We appreciate that the reviewer found satisfactory the new data on the neutralizing antibodies titers to ZIKV at 30 and 60 days after ZIKV infection. We added a comment on the difference replication capabilities of these two strains (Lines 533-538):

"These results suggest that the differences in the neutralization profile between the two ZIKV-immune groups are associated to the longevity of ZIKV convalescence which may be attributable to the maturation of the cross-reactive immune memory elicited by the heterologous DENV infection and no to the antigenic differences or the different replication capabilities in rhesus macaques of those two pre-infecting ZIKV strains^{17,54}."

In the Methods section, we now mentioned that the ZIKVPR-2mo cohort was challenged separately (Lines 775-777):

"Cohort 1 and 3 were challenged on the same day while cohort 2 was challenged 3 months later with the same stock of DENV-2. However, all samples were frozen and analyzed together, except for the immunophenotyping analysis."

We appreciate that the reviewer found acceptable the reasons provided supporting our rationale that the challenge of one cohort at a different time of the other two do not explain differences between groups. However, providing all those arguments in the manuscript will result in an extensive discussion.

As we explained before, challenging groups of NHPs or even mice at different time points is a general practice. Particularly when an experiment needs to be repeated to add more data or to confirm previous results. We do believe that with the clarifications provided following the reviewers' suggestions, the scientific readers will agree on and understand the schedule of this experiment.

3. Statistical analyses- The authors appear to have missed the point that reporting p value estimations to 4 significant digits suggests a precision that is not typically considered appropriate for such small samples. Their comments (in the rebuttal and the discussion) citing other papers using similar group sizes are not pertinent to this point, but I would yield to the usual practice of the journal. Additional points regarding specific figures:

We agree with the reviewer about the 4 significant digits vs the small sample. We choose to yield to the usual practice of this journal as well and let the Editors decide on this particular point. However, as the reviewer acknowledged we are providing similar statistical interpretations that have been done in previous works in different journals (including NatComms) using a similar or smaller number of animals.

Figure 2a- The meaning of the box and whiskers should be specified.

The meaning for both is now specified in the figure legend of Figure 2.

Figure 2b- There is no information on the distribution of values within each group.

The figure was modified and the distribution for individual animals per cohort is now shown.

Figure S2- The Methods do not indicate what value was imputed for samples below LOD in calculation of means.

The value assigned is now provided in the figure legend of Supplementary Figure 2.

4. Limitations of the study- I appreciate the authors' effort to address this point. A point that should be considered is that the study only included two specific time points after ZIKV infection, 2 and 10 months. The authors do not address the appropriateness of selecting these two time points and the potential for different findings at other biologically relevant intervals.

We appreciate the reviewer's observation. We added the following comment (Lines 141-143):

"The 2 months cohort was selected for direct comparison with previous work in NHP³⁸, while the 10 months cohort was selected based on availability to test a longer period of convalescence to ZIKV."

5. Viremia data- I appreciate the authors' clarifications. Regarding the new analysis based on AUC, it would be more informative to calculate this value separately for each animal and compare groups using appropriate statistical tests. (In this case, the authors should be sure to note that this was a post-hoc analysis.)

We appreciate that the reviewer found the clarifications appropriated. Reviewer 1 suggested the AUC analysis and we completed it. We believe that the AUC analysis added supportive data on the effect of previous ZIKV immunity on DENV replication kinetics. We respect Reviewer 3 point of view on that analysis. But we believe that adding a new analysis, calculating the AUC for each individual will no substantially expand the results we are presenting.

6. Cytokine levels- I believe it would be helpful to the readers for the authors to have noted which specific cytokines were found elevated in the study by George et al.

While we agree with the reviewer, we think that mentioning the 9 pro-inflammatory cytokines that George et al. claimed they found significantly elevated will be repetitive and will result in an extended discussion. On the other hand, we are including another important factor that may be responsible for these discrepancies in the cytokine profile between both studies. For this reason, we may not provide a direct comparison with their findings on this aspect. We added the following sentences (Lines: 655-658):

"Contrary to our findings, a previously published work found that an approximately two month ZIKV immunity period resulted in an increase of pro-inflammatory cytokines³⁸. However, a differential effect due to the use of different sample types (plasma vs serum) between both studies cannot be ruled out."

7. Figure S13 (formerly S10)- The authors response does not allay my specific point of concern. Comparing the data for the ZIKVPR-2mo group between panels d, e, and f, most values for the percentages of CD69+ cells in total T cells appear to be <5%, whereas corresponding values for CD4 and CD8 subsets often range over 20%. The authors were directed to these findings specifically because normally these two subsets would encompass nearly all T cells. The authors' explanation wouldn't be adequate unless there was a substantial percentage of T cells that did not fall into either subset. Was that the case, and, if so,

what is the authors' explanation?

We are sorry that our initial answer did not cover the reviewer concern. We appreciate the opportunity to provide a more specific answer.

The reason for this apparent anomaly between the % of CD69+ T cells and the corresponding % of CD69+ CD4 and CD69+ CD8 T cells for the same sample is due to the fact that the graphs are presented as the percentage values of CD69+ per total cell population (T cells) or subpopulation (CD4 T or CD8 T cells). For example, a sample may show 8.5 % of CD69+ T cells, 3.5% of CD69+ CD4 T cells, and 18% of CD69+ CD8 T cells, with total T cells for this sample consisting of 65% of CD4 T cells and 35% of CD8 T cells. This means that the contribution of CD69+ CD4 T cells to the total T cell population is $3.5 \times 0.65 = 2.3$, while for CD69+ CD8 T cells we obtain $18 \times 0.35 = 6.3$. These CD69+ CD4 T and CD69+ CD8 T values combined ($2.3 + 6.3 = 8.6$) are similar to the 8.5% of CD69+ total T cells reported.

However, this data is not considered essential to support the main findings of this work. Because of that and to avoid further confusion we decided to eliminate panels d-f from the Supplementary Figure 13. Also, we removed the text that refers to these panels:

“Moreover, the overall T cell response was activated by DENV infection in all groups (Supplementary Fig. 13d-f).”